# ATG6 interacting with NPR1 increases *Arabidopsis thaliana* resistance to *Pst* DC3000/*avrRps4* by increasing its nuclear accumulation and stability

Baihong Zhang[1], Shuqin Huang[1], Shuyu Guo[2,3], Yixuan Meng[1], Yuzhen Tian[1], Yue Zhou[1], Hang Chen[1], Xue Li[1], Jun Zhou[1]*, Wenli Chen[1]*

[1]MOE Key Laboratory of Laser Life Science & Institute of Laser Life Science, Guangdong Provincial Key Laboratory of Laser Life Science, Guangzhou Key Laboratory of Spectral Analysis and Functional Probes, College of Biophotonics, School of Optoelectronic Science and Engineering, South China Normal University, Guangzhou, China; [2]State Key Laboratory of Reproductive Regulation and Breeding of Grassland Livestock, College of Life Sciences, Inner Mongolia University, Hohhot, China; [3]Key Laboratory of Herbage and Endemic Crop Biotechnology, and College of Life Sciences, Inner Mongolia University, Hohhot, China

*For correspondence:
zhoujun@scnu.edu.cn (JZ);
chenwl@scnu.edu.cn (WC)

## eLife Assessment

This **important** study investigates the role of ATG6 in regulating NPR1, a key protein in the plant immune response. The authors present **compelling** evidence that ATG6 not only interacts with NPR1 in both the cytoplasm and nucleus but also enhances its stability and nuclear accumulation, leading to increased resistance to Pst DC3000/avrRps4 infection in *Arabidopsis thaliana*. The work incorporates a variety of approaches from molecular biology, confocal imaging, and biochemistry, which together strengthen the conclusions.

**Abstract** Autophagy-related gene 6 (ATG6) plays a crucial role in plant immunity. Nonexpressor of pathogenesis-related genes 1 (NPR1) acts as a signaling hub of plant immunity. However, the relationship between ATG6 and NPR1 is unclear. Here, we find that ATG6 directly interacts with NPR1. *ATG6* overexpression significantly increased nuclear accumulation of NPR1. Furthermore, we demonstrate that *ATG6* increases NPR1 protein levels and improves its stability. Interestingly, ATG6 promotes the formation of SINCs (SA-induced NPR1 condensates)-like condensates. Additionally, ATG6 and NPR1 synergistically promote the expression of *pathogenesis-related* genes. Further results showed that silencing *ATG6* in *NPR1-GFP* exacerbates *Pst* DC3000/*avrRps4* infection, while double overexpression of *ATG6* and *NPR1* synergistically inhibits *Pst* DC3000/*avrRps4* infection. In summary, our findings unveil an interplay of NPR1 with ATG6 and elucidate important molecular mechanisms for enhancing plant immunity.

## Introduction

Plants are constantly challenged by pathogens in nature. In order to survive and reproduce, plants have evolved complex mechanisms to cope with attack by pathogens (*Jones and Dangl, 2006*). Nonexpressor of pathogenesis-related genes 1 (NPR1) is a key regulator of plant immunity (*Chen*

et al., 2021b). It contains the BTB/POZ (Broad Compex, Tramtrack, and BricaBrac/Pox virus and Zinc finger) domain in the N-terminal region, the ANK (Ankyrin repeats) domain in the middle region, and SA-binding domain (SBD) and the nuclear localization sequence (NLS) in the C-terminal region (**Cao et al., 1997**; **Rochon et al., 2006**; **Kumar et al., 2022**). NPR1 is a receptor of SA (salicylic acid) mainly localized as an oligomer in the cytoplasm and sensitive to the surrounding redox state (**Tada et al., 2008**; **Wu et al., 2012**). SA mediates the dynamic oligomer to dimer response of NPR1 (**Tada et al., 2008**) and promotes translocation of NPR1 into the nucleus, which increases plant resistance to pathogens by activating the expression of immune-related genes (**Kinkema et al., 2000**; **Chen et al., 2021b**).

NPR1 is mainly degraded by the ubiquitin proteasome system (UPS) (**Spoel et al., 2009**; **Saleh et al., 2015**; **Skelly et al., 2019**). An increasing researches have shown that autophagy and the UPS pathway play overlapping roles in regulating intracellular protein homeostasis (**Zhou et al., 2014**; **Marshall et al., 2015**; **Kikuchi et al., 2020**). Our previous study showed that ATGs (autophagy-related genes) are involved in NPR1 turnover (**Gong et al., 2020**). Autophagy negatively regulates *Pst* DC3000/*avrRpm1*-induced programmed cell death (PCD) via the SA receptor NPR1 (**Yoshimoto et al., 2009**). These results imply that ATGs might be involved in plant immunity through the regulation of NPR1 homeostasis. However, the detailed mechanism has not yet been elucidated.

ATG6 is the homologs of yeast Vps30/Atg6 and mammalian BECN1/Beclin1 (**Xu et al., 2017**). It is a common and required subunit of the class III phosphatidylinositol 3-kinase (PtdIns3K) lipid kinase complexes, which regulates autophagosome nucleation in *Arabidopsis thaliana* (*Arabidopsis*) (**Qi et al., 2017**; **Wang et al., 2020**). The homozygous *atg6* mutant is lethal, suggesting that ATG6 is essential for plant growth and development (**Fujiki et al., 2007**; **Qin et al., 2007**; **Harrison-Lowe and Olsen, 2008**; **Patel and Dinesh-Kumar, 2008**). In *Arabidopsis*, *Nicotiana benthamiana* and wheat, ATG6 or its homologs was reported to act as a positive regulator to enhance plant disease resistance to *P. syringae pv. tomato* (*Pst*) DC3000 and *Pst* DC3000/*avrRpm1* bacteria (**Patel and Dinesh-Kumar, 2008**), *N. benthamiana* mosaic virus (TMV) (**Liu et al., 2005**), turnip mosaic virus (TuMV) (**Li et al., 2018**), pepper mild mottle virus (PMMoV) (**Jiao et al., 2020**), and *Blumeria graminis f. sp. tritici* (*Bgt*) fungus (**Yue et al., 2015**). Several research teams have also elucidated that ATG6 interacted with Bax Inhibitor-1 (NbBI-1) **Xu et al., 2017** and RNA-dependent RNA polymerase (RdRp) (**Li et al., 2018**) to suppress pathogen infection. However, the mechanism by which ATG6 suppresses pathogen infection by regulating NPR1 has not yet been reported.

Here, we show that ATG6 and NPR1 synergistically enhance *Arabidopsis* resistance to *Pst* DC3000/*avrRps4* infiltration. We discover that ATG6 increases NPR1 protein levels and nuclear accumulation of NPR1. Moreover, ATG6 can stabilize NPR1 and promote the formation of SINCs (SA-induced NPR1 condensates)-like condensates. Our study revealed a unique mechanism in which NPR1 cooperatively increases plant immunity with ATG6.

## Results

### NPR1 physically interacts with ATG6 in vitro and in vivo

To examine the relationship between ATGs and NPRs, we predicted that some ATGs might interact with NPRs. In a yeast two-hybrid (Y2H) screen, we identified that NPR1, NPR3, and NPR4 could interact with ATG6 and several ATG8 isoforms (**Figure 1—figure supplement 1** and Appendix 1—result 1). In this study, we mainly investigated the relationship between ATG6 and NPR1 during the process of plant immune response. First, the NPR1 truncations NPR1-N (1-328AA, containing the BTB/POZ domain, ANK1, ANK2) and NPR1-C (328-594AA, containing the ANK3, ANK4, SBD, and NLS) were used to identify the interaction domains between NPR1 and ATG6. The results showed that NPR1-C interacted with full-length ATG6 in yeast (**Figure 1a**, line 3). The interaction between NPR1 and SnRK2.8 was used as a positive control (**Lee et al., 2015**). Second, pull-down assays were performed in vitro. NPR1-His bound GST-ATG6, but not GST (**Figure 1b**). Furthermore, co-immunoprecipitation assays were performed in *N. benthamiana*, as shown in **Figure 1c**, ATG6-mCherry was co-immunoprecipitated with NPR1-GFP. In **Figure 1—figure supplement 2**, fluorescence signals of NPR1-GFP and ATG6-mCherry were co-localized in both the nucleus and cytoplasm. The interaction between ATG6 and NPR1 was also verified by a bimolecular fluorescence complementation (BiFC) assay (**Figure 1d, e**). These results demonstrate that ATG6 interacts with NPR1 both in vitro and in vivo.

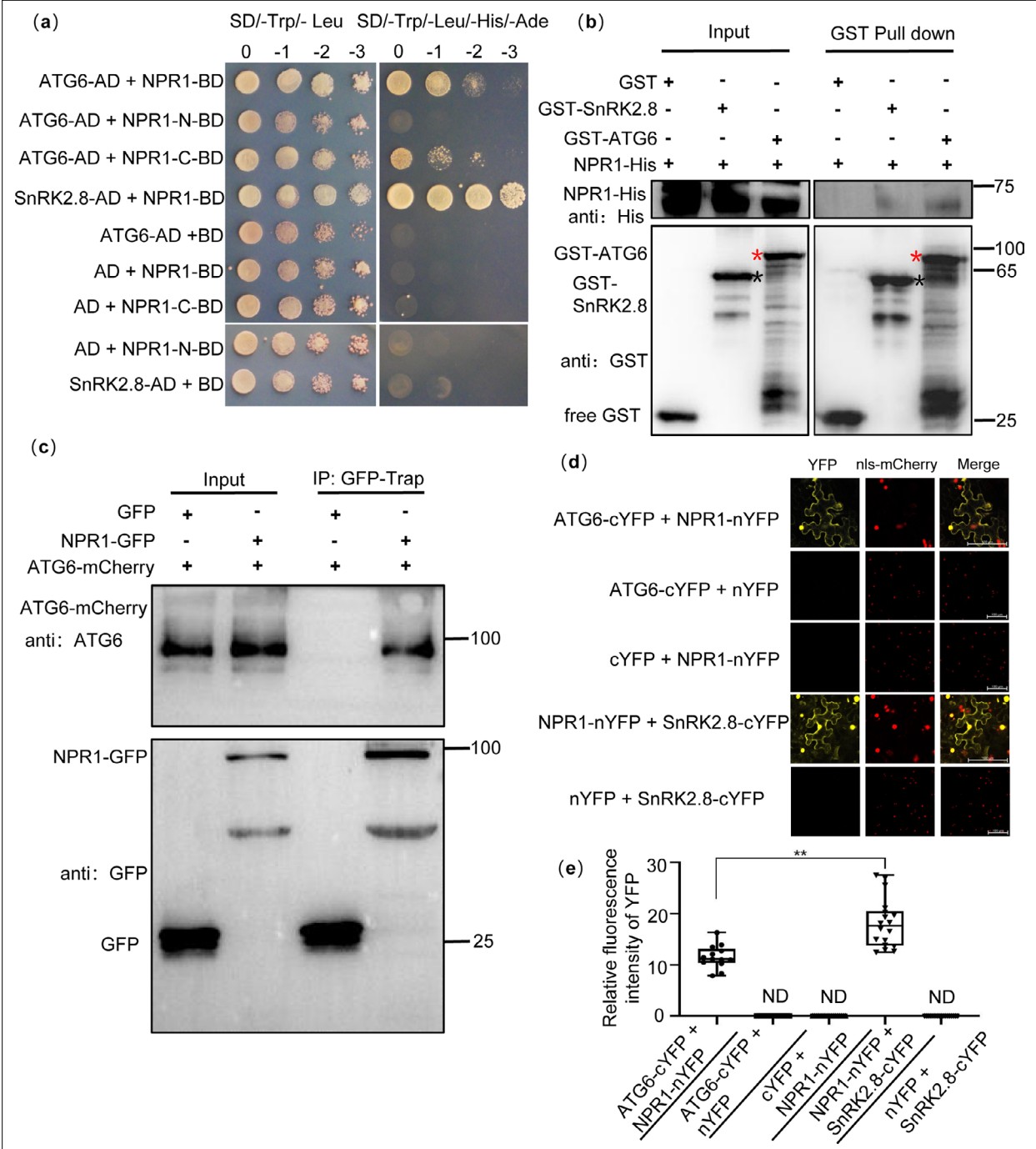

**Figure 1.** Physical interaction between NPR1 and ATG6. (**a**) Interaction of NPR1 with ATG6 in yeast. The CDS of *ATG6*, *NPR1*, *NPR1-N* (1–984 bp), *NPR1-C* (984–1782 bp), and *SnRK2.8* were fused to pGADT7 (AD) and pGBKT7 (BD), respectively. Co-transformation of NPR1-BD + AD, BD + ATG6-AD, BD + SnRK2.8-AD, NPR1-N-BD + AD, and NPR1-C-BD + AD were used as negative controls. The interaction of NPR1-BD and SnRK2.8-AD was used as a positive control. Yeast growth on SD/-Trp-Leu-His-Ade media represents interaction. Numbers represent the dilution fold of yeast. 0, –1 (10-fold dilution), –2 (100-fold dilution), and –3 (1000-fold dilution). (**b**) In vitro pull-down assays of NPR1-His with GST-ATG6 fusion protein. NPR1-His prokaryotic proteins were incubated with GST-tag Purification Resin conjugated with GST-ATG6, GST, and SnRK2.8-GST. Western blotting analysis with anti-GST and anti-His. Black asterisk indicates SnRK2.8-GST bands. Red asterisk indicates GST-ATG6 bands. (**c**) Co-immunoprecipitation of NPR1 with ATG6 in vivo. Total protein was extracted from *Nicotiana benthamiana* transiently transformed with ATG6-mCherry + GFP and ATG6-mCherry + NPR1-GFP, followed by IP with GFP-Trap. Western blots analysis with ATG6 and GFP antibodies. (**d**) Bimolecular fluorescence complementation assay of NPR1 with ATG6 in *N. benthamiana* leaves. Agrobacterium carrying ATG6-cYFP and NPR1-nYFP was co-expressed in leaves of *N. benthamiana* for 3 days. As a positive control, NPR1-nYFP and SnRK2.8-cYFP were co-expressed. As negative controls, nYFP and ATG6-cYFP, NPR1-nYFP and cYFP, nYFP and

*Figure 1 continued on next page*

*Figure 1 continued*

SnRK2.8-cYFP were co-expressed. Confocal images were obtained from mCherry, YFP. nls-mCherry as a nuclear localization mark. Scale bar = 100 µm. (**e**) Relative fluorescence intensity of YFP in (**d**) using ImageJ software, ND means not detected, *n* = 15 independent images were analyzed to quantify YFP fluorescence. ** indicates that the significant difference compared to the control is at the level of 0.01 (Student's *t*-test p value, **p < 0.01). All experiments were performed with three biological replicates.

The online version of this article includes the following source data and figure supplement(s) for figure 1:

**Source data 1.** Original files for western blot analysis displayed in *Figure 1b, c*.

**Source data 2.** PDF file containing original western blots for *Figure 1b, c*, indicating the relevant bands and treatments.

**Source data 3.** Numerical source data files for *Figure 1e*.

**Figure supplement 1.** Physical interaction between NPRs and ATGs in yeast.

**Figure supplement 2.** Co-localization of NPR1-GFP and ATG6-mCherry in *N. benthamiana*.

## ATG6 co-localized with NPR1 in the nucleus

Remarkably, we found that ATG6 is localized in the cytoplasm and nucleus, and it co-localized with NPR1 in the nucleus (*Figure 1—figure supplement 2*). Nuclear localization of ATG6 was also observed in *N. benthamiana* transiently transformed with ATG6-mCherry and ATG6-GFP under normal and SA treatment conditions (*Figure 2a, b*). ATG6-GFP co-localizes with the nuclear localization marker nls-mCherry (indicated by white arrows) (*Figure 2b*). Additionally, we observed punctate patterns indicative of canonical autophagy-like localization of ATG6-GFP fluorescence signals (indicated by red circles) (*Figure 2b*). The nuclear localization signal of ATG6 was also observed in *UBQ10::ATG6-GFP* overexpressing *Arabidopsis* (*Figure 2—figure supplement 1a*). To exclude the possibility that the observed localization of ATG6-GFP is due to free GFP. The protein levels of ATG6-GFP and free GFP in *UBQ10::ATG6-GFP Arabidopsis* and *N. benthamiana* were detected before and after SA treatment. Notably, no free GFP was detected and this means that the fluorescence signal observed by confocal microscopy is ATG6-GFP, not free GFP (*Figure 2—figure supplement 1d, e*). In both plants and animals, proteins are transported to the nucleus via specific nuclear localization signals (NLSs), which are typically characterized by short stretches of basic amino acids (*Dingwall and Laskey, 1991*; *Raikhel, 1992*; *Nigg, 1997*). Furthermore, we analyzed the putative NLS in the ATG6 protein sequence using NLSExplorer (http://www.csbio.sjtu.edu.cn/bioinf/NLSExplorer). Although we did not identify a classical monopartite NLS, we discovered a bipartite NLS similar to the consensus bipartite sequence $(KRX_{(10-12)}K(KR)(KR))$ (*Kosugi et al., 2009*) in the carboxy-terminal region (475–517 aa) of ATG6, with a cut-off score of 2.6 (*Figure 2c*). Additionally, our comparison of ATG6 C-terminal sequences across several species, including *Microthlaspi erraticum*, *Capsella rubella*, *Brassica carinata*, *Camelina sativa*, *Theobroma cacao*, *Brassica rapa*, *Eutrema salsugineum*, *Raphanus sativus*, *Hirschfeldia incana*, and *Brassica napus*, sequence comparison indicates that this bipartite NLS is relatively conserved (*Figure 2c*).

Moreover, the nuclear and cytoplasmic fractions were separated. Under SA treatment, ATG6-mCherry and ATG6-GFP were detected in the cytoplasmic and nuclear fractions in *N. benthamiana* (*Figure 2d, e*). However, in *N. benthamiana*, we observed that ATG6-mCherry was not detected in the nuclear fractions under normal conditions, which differents with the results shown in *Figure 2a*. We suspect that this discrepancy may be due to the fluorescence signal in *Figure 2a* primarily arising from free mCherry rather than the ATG6-mCherry fusion. ATG6 was also detected in the nuclear fraction of *UBQ10::ATG6-GFP* and *UBQ10::ATG6-mCherry* overexpressing plants, and SA promoted both cytoplasm and nuclear accumulation of ATG6 (*Figure 2f, g*). Additionally, we obtained *ATG6* and *NPR1* double overexpression of *Arabidopsis UBQ10::ATG6-mCherry × 35S::NPR1-GFP* (*ATG6-mCherry × NPR1-GFP*) by crossing and screening (*Figure 2—figure supplement 2a*). In *ATG6-mCherry × NPR1-GFP*, we observed co-localization of ATG6-mCherry with NPR1-GFP in the nucleus (*Figure 2—figure supplement 1b*). These results are consistent with the prediction of the subcellular location of ATG6 in the *Arabidopsis* subcellular database (https://suba.live/) (*Figure 2—figure supplement 1c*). Additionally, we have conducted an investigation into the localization of endogenous ATG6 in Col. Our results demonstrate that endogenous ATG6 is present in both the nucleus and cytoplasm, and we have observed that SA treatment promotes the accumulation of ATG6 in the nucleus (*Figure 2—figure supplement 3*). Together, these findings suggest that ATG6 is localized to both cytoplasm and nucleus, and co-localized with NPR1 in the nucleus.

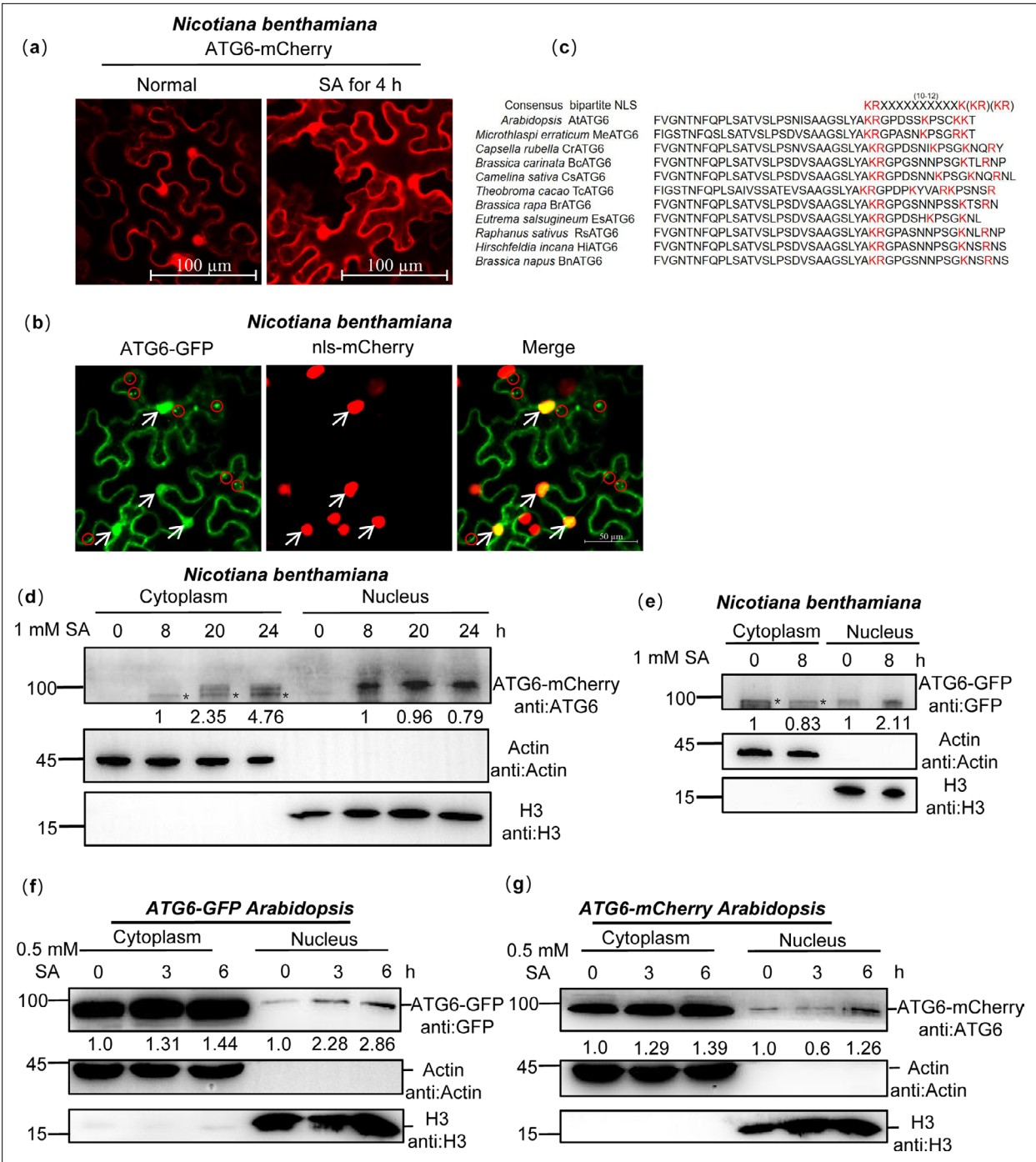

**Figure 2.** ATG6 is localized in the cytoplasm and nucleus. (**a**) The nuclear localization of ATG6-mCherry in *N. benthamiana*. Scale bar, 100 μm. (**b**) Co-localization of ATG6-GFP and nls-mCherry in *N. benthamiana*. Scale bar, 50 μm. (**c**) ATGs protein nuclear localization sequence analysis using the online NLSExplorer prediction software and sequence comparison of ATG6 C-terminal with other species. (**d**) Subcellular fractionation of ATG6-mCherry in *N. benthamiana* after 1 mM SA treatment. Black asterisk (*) indicates ATG6-mCherry bands. (**e**) Subcellular fractionation of ATG6-GFP in *N. benthamiana* after 1 mM SA treatment. Black asterisk (*) indicates ATG6-GFP bands. (**f**) Subcellular fractionation of ATG6-GFP in *ATG6-GFP Arabidopsis* after 0.5 mM SA treatment. (**g**) Subcellular fractionation of ATG6-mCherry in *ATG6-mCherry Arabidopsis* after 0.5 mM SA treatment. In (**d–g**), ATG6-mCherry (**d, g**) and ATG6-GFP (**e, f**) were detected using ATG6 or GFP antibody. Actin and H3 were used as cytoplasmic and nucleus internal reference, respectively. Numerical values indicate quantitative analysis of ATG6-mCherry and ATG6-GFP using ImageJ. All experiments were performed with three biological replicates.

The online version of this article includes the following source data and figure supplement(s) for figure 2:

**Source data 1.** Original files for western blot analysis displayed in *Figure 2d–g*.

*Figure 2 continued on next page*

*Figure 2 continued*

**Source data 2.** PDF file containing original western blots for *Figure 2d–g*, indicating the relevant bands and treatments.

**Figure supplement 1.** The nuclear localization of ATG6 in *Arabidopsis*.

**Figure supplement 1—source data 1.** Original files for western blot analysis displayed in *Figure 2—figure supplement 1d, e*.

**Figure supplement 1—source data 2.** PDF file containing original western blots for *Figure 2—figure supplement 1d, e*, indicating the relevant bands and treatments.

**Figure supplement 2.** Identification of *ATG6-mCherry × NPR1-GFP* plants.

**Figure supplement 2—source data 1.** Original files for western blot analysis displayed in *Figure 2—figure supplement 2a, b*.

**Figure supplement 2—source data 2.** PDF file containing original western blots for *Figure 2—figure supplement 2a, b*, indicating the relevant bands and treatments.

**Figure supplement 3.** Subcellular fractionation of endogenous ATG6 in Col after 0.5 mM SA treatment for 0, 3, 6, and 24 hr.

**Figure supplement 3—source data 1.** Original files for western blot analysis displayed in *Figure 2—figure supplement 3*.

**Figure supplement 3—source data 2.** PDF file containing original western blots for *Figure 2—figure supplement 3*, indicating the relevant bands and treatments.

## ATG6 overexpression increased nuclear accumulation of NPR1

Previous studies have shown that the nuclear localization of NPR1 is essential for improving plant immunity (*Kinkema et al., 2000*; *Chen et al., 2021b*). We observed that a stronger nuclear localization signal of NPR1-GFP in *ATG6-mCherry × NPR1-GFP* leaves than that in *NPR1-GFP* under normal condition and 0.5 mM SA treatment for 3 hr (*Figure 3a, b* and *Figure 3—figure supplement 1*). These findings indicate that ATG6 might increase nuclear accumulation of NPR1. To exclude the possibility that the observed localization of NPR1-GFP is due to free GFP, we detected the levels of NPR1-GFP and free GFP in *ATG6-mCherry × NPR1-GFP* plants before and after SA treatment. Only ~10% of free GFP was detected in *ATG6-mCherry × NPR1-GFP* plants before and after SA treatment, confirming that the observed localization of NPR1-GFP is not due to free GFP (*Figure 2—figure supplement 2b*). Furthermore, the nuclear and cytoplasmic fractions of *ATG6-mCherry × NPR1-GFP* and *NPR1-GFP* were separated. Under normal conditions, the nuclear fractions NPR1-GFP in *ATG6-mCherry × NPR1-GFP* and *NPR1-GFP* were relatively weaker (*Figure 3c*), which differs from the above observation (*Figure 3a*). We speculate that this phenomenon might be attributed to the rapid turnover of NPR1 in the nucleus (*Spoel et al., 2009*; *Saleh et al., 2015*). Consistent with the fluorescence distribution results, the nuclear fractions of NPR1-GFP in *ATG6-mCherry × NPR1-GFP* were significantly higher than those in *NPR1-GFP* under 0.5 mM SA treatment for 3 and 6 hr (*Figure 3c* and *Figure 3—figure supplement 2*). Furthermore, *Agrobacterium* harboring ATG6-mCherry and NPR1-GFP were transiently transformed to *N. benthamiana* leaves. After 1 day, the leaves were treated with 1 mM SA for 8 and 20 hr. Subsequently nucleoplasmic separation experiments were performed. Similar to *Arabidopsis*, increased nuclear accumulation of NPR1 was found when *ATG6* was overexpressed (*Figure 3e* and *Figure 3—figure supplement 2*). Notably, we found that the ratio (nucleus NPR1/total NPR1) in *ATG6-mCherry × NPR1-GFP* was not significantly different from that in *NPR1-GFP* after SA treatment, and a similar phenomenon was observed in *N. benthamiana* (*Figure 3d, f* and *Figure 3—figure supplement 2*). These results suggested that the increased nuclear accumulation of NPR1 in *ATG6-mCherry × NPR1-GFP* plants might attributed to higher levels and more stable NPR1 rather than the enhanced nuclear translocation of NPR1 facilitated by ATG6. Furthermore, we validated the functionality of the ATG6-GFP and ATG6-mCherry fusion proteins utilized in this study by examining the phenotypes of *ATG6-GFP* and *ATG6-mCherry Arabidopsis* plants under carbon starvation conditions (*Figure 3—figure supplement 3* and Appendix 1—result 2).

## ATG6 increases endogenous SA levels and promotes the expression of NPR1 downstream target genes

NPR1 localized in the nucleus is essential for activation of immune gene expression (*Kinkema et al., 2000*; *Chen et al., 2021b*). In our study, we observed that *ATG6* overexpression increased nuclear accumulation of NPR1 (*Figure 3*) and demonstrated an interaction between ATG6 and NPR1 in the nucleus (*Figure 1d*). Therefore, we speculate that ATG6 might regulate NPR1 transcriptional activity. Notably, the expression level of *ICS1* in *ATG6-mCherry × NPR1-GFP* seedlings was significantly higher

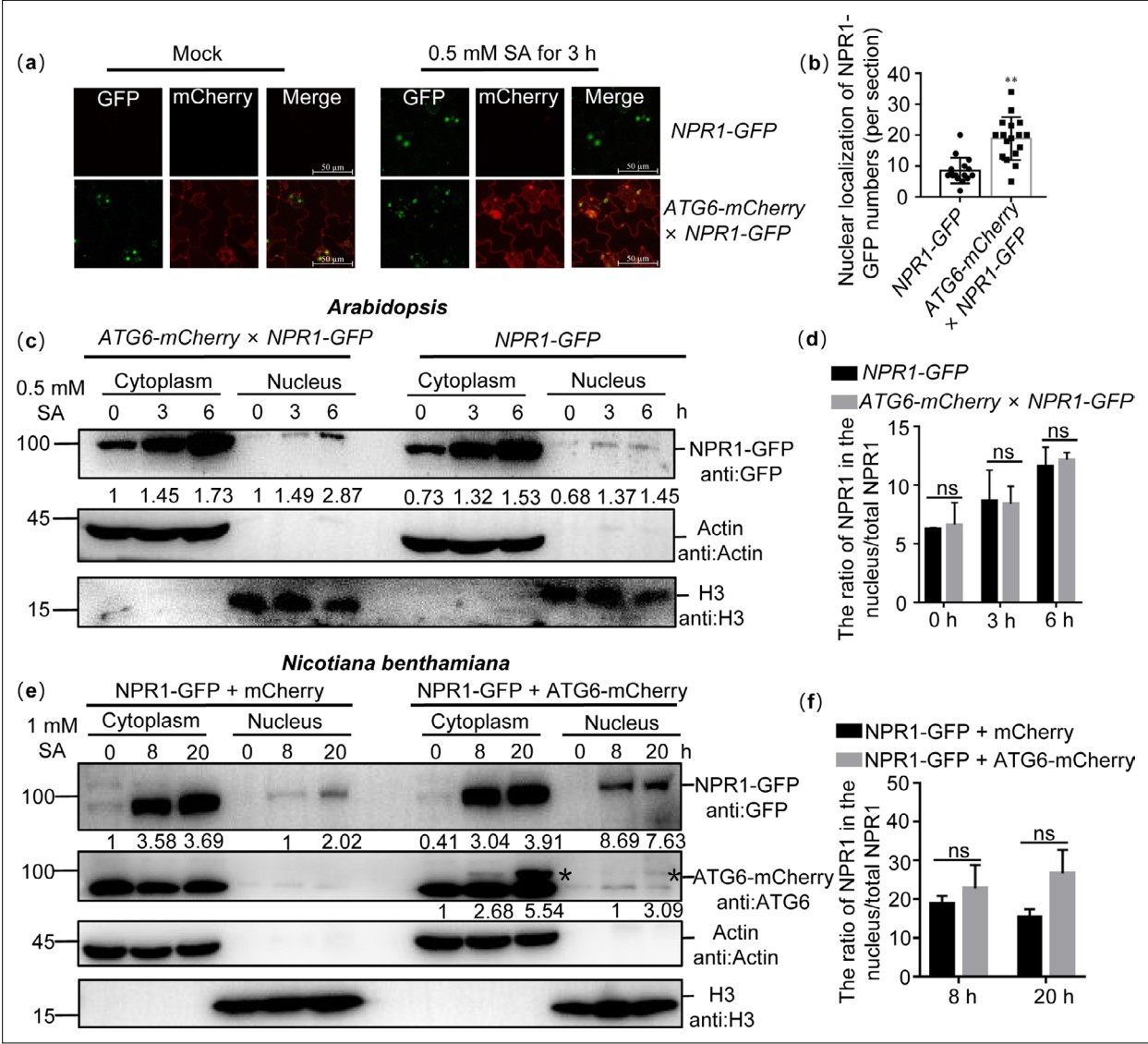

**Figure 3.** ATG6 increases the nuclear accumulation of NPR1 under SA treatment. (**a**) Confocal images of NPR1-GFP nuclear localization in 7-day-old seedlings of *NPR1-GFP* and *ATG6-mCherry × NPR1-GFP* under normal and 0.5 mM SA spray for 3 hr. Scale bar, 50 μm. (**b**) The count of nuclear localizations of NPR1-GFP in *ATG6-mCherry × NPR1-GFP* and *NPR1-GFP Arabidopsis* plants following SA treatment in (**a**). Statistical data were obtained from three independent experiments, each comprising five individual images, resulting in a total of 15 images analyzed for this comparison. ** indicates that the significant difference compared to the control is at the level of 0.01 (Student's *t*-test p value, **p < 0.01). (**c**) Subcellular fractionation of NPR1-GFP in 7-day-old seedlings of *NPR1-GFP* and *ATG6-mCherry × NPR1-GFP* after 0.5 mM SA treatment for 0, 3, and 6 hr. (**d**) The ration of NPR1 in the nucleus/total NPR1 in (**c**), Student's *t*-test was conducted to analyze the data. The mean and standard deviation were calculated from three biological replicates, ns indicates no significant difference. (**e**) Subcellular fractionation of NPR1-GFP in *N. benthamiana* after 1 mM SA treatment for 0, 8, and 20 hr. (**f**) The ration of NPR1 in the nucleus/total NPR1 in (**e**), Student's *t*-test was conducted to analyze the data. The mean and standard deviation were calculated from three biological replicates, ns indicates no significant difference. In (**c, e**), cytoplasmic and nuclear proteins were extracted from *Arabidopsis* or *N. benthamiana*. NPR1-GFP were detected using GFP antibody. Actin and H3 were used as cytoplasmic and nucleus internal reference, respectively. Numerical values indicate quantitative analysis of NPR1-GFP using ImageJ. All experiments were performed with three biological replicates.

The online version of this article includes the following source data and figure supplement(s) for figure 3:

**Source data 1.** Original files for western blot analysis displayed in *Figure 3c, e*.

**Source data 2.** PDF file containing original western blots for *Figure 3c, e*, indicating the relevant bands and treatments.

**Source data 3.** Numerical source data files for *Figure 3b, d, f*.

**Figure supplement 1.** Confocal images of NPR1-GFP nuclear localization in 7-day-old seedlings of *NPR1-GFP* and *ATG6-mCherry × NPR1 -GFP* under normal and 0.5 mM SA spray for 3 hr.

*Figure 3 continued on next page*

than that in *NPR1-GFP* under normal and SA treatment conditions (*Figure 4—figure supplement 1*). Free SA levels in *ATG6-mCherry × NPR1-GFP* were also significantly higher compared to *NPR1-GFP* under *Pst* DC3000/*avrRps4* treatment. While there was no significant difference was observed under normal condition (*Figure 4a*), this may be related to free SA consumption, as it can be converted to bound SA (*Ding and Ding, 2020*). In addition, the expression of *PR1* (*pathogenesis-related gene 1*) and *PR5* in *ATG6-mCherry × NPR1-GFP* was significantly higher than that of *NPR1-GFP* under normal and SA treatment conditions (*Figure 4b, c*). The expression of *PR1* and *PR5* in *ATG6-mCherry* was significantly higher than that of Col under *Pst* DC3000/*avrRps4* treatment (*Figure 4—figure supplement 2*). These results support the role of ATG6 in facilitating the expression of NPR1 downstream *PR1* and *PR5* genes.

## ATG6 increases NPR1 protein levels and the formation of SINCs-like condensates

Interestingly, similar to previous reports (*Zavaliev et al., 2020*), SA promoted the translocation of NPR1 into the nucleus, but still a significant amount of NPR1 was present in the cytoplasm (*Figure 3c, e*). Previous studies have shown that SA increased NPR1 protein levels and facilitated the formation of SINCs in the cytoplasm, which are known to promote cell survival (*Zavaliev et al., 2020*). In our experiments, we observed that under SA treatment, the protein levels of NPR1 in *ATG6-mCherry ×*

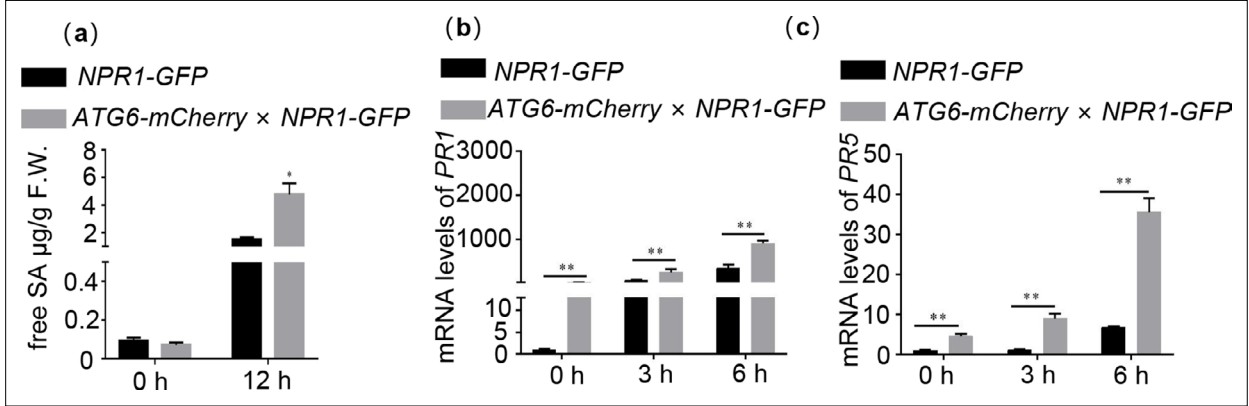

**Figure 4.** ATG6 increases endogenous SA levels and promotes the expression of NPR1 downstream target genes. (**a**) Level of free SA in 3-week-old *NPR1-GFP* and *ATG6-mCherry × NPR1-GFP* after *Pst* DC3000/*avrRps4* for 12 hr. Expression of *PR1* (**b**) and *PR5* (**c**) in 3-week-old *NPR1-GFP* and *ATG6-mCherry × NPR1-GFP* under normal and SA treatment conditions, values are means ± SD (*n* = 3 biological replicates). The *AtActin* gene was used as the internal control. * or ** indicates that the significant difference compared to the control is at the level of 0.05 or 0.01 (Student's *t*-test p value, *p < 0.05 or **p < 0.01). All experiments were performed with three biological replicates.

The online version of this article includes the following source data and figure supplement(s) for figure 4:

**Source data 1.** Numerical source data files for *Figure 4a–c*.

**Figure supplement 1.** Expression of *ICS1* in 3-week-old *NPR1-GFP* and *ATG6-mCherry × NPR1-GFP* under normal and 0.5 mM SA treatment conditions.

**Figure supplement 1—source data 1.** Numerical source data files for *Figure 4—figure supplement 1*.

**Figure supplement 2.** Expression of *PR1* and *PR5* in Col and *ATG6-mCherry* under normal and *Pst* DC3000/*avrRps4* treatment.

**Figure supplement 2—source data 1.** Numerical source data files for *Figure 4—figure supplement 2*.

**Figure supplement 3.** Structural analysis of acidic activation domains (AADs) in ATG6. Acidic (red) and hydrophobic (blue) amino acid residues in AADs.

*NPR1-GFP* was significantly higher than that in *NPR1-GFP* (*Figure 5a*). To further support our conclusions, we proceeded to silence *ATG6* in *NPR1-GFP* (*NPR1-GFP*/silencing *ATG6*) and subsequently assessed the protein level of NPR1-GFP before and after SA treatment. Our findings revealed that the protein level of NPR1-GFP in *NPR1-GFP*/silencing *ATG6* under SA treatment was notably lower than that in the *NPR1-GFP*/Negative control (*Figure 5—figure supplement 1*). Under SA treatment for 8 hr, the protein levels of NPR1-GFP in *N. benthamiana* co-transformed with ATG6-mCherry + NPR1-GFP was also significantly higher than that of mCherry + NPR1-GFP (*Figure 5b*). While there was a slight increase at 20 hr, a minor decrease was observed at 24 hr, suggesting that the rise in NPR1 protein levels induced by ATG6 was transient. We also detected the expression of *NPR1* was detected. It is worth noting that NPR1 up-regulation was more obvious in Col after 3 hr treatment with *Pst* DC3000/*avrRps4*. After 6 hr treatment with *Pst* DC3000/*avrRps4*, there was no significant difference in the expression of *NPR1* between Col and *ATG6-mCherry* (*Figure 5—figure supplement 2*). These results suggest that ATG6 increases NPR1 protein levels. After SA treatment, more SINCs-like condensates fluorescence were observed in *N. benthamiana* co-transformed with ATG6-mCherry + NPR1-GFP compared to mCherry + NPR1-GFP (*Figure 5c, d*, *Videos 1 and 2*). Additionally, we observed that SINCs-like condensates signaling partial co-localized with certain ATG6-mCherry autophagosomes fluorescence signals (*Figure 5—figure supplement 3*). Taken together, these results suggest that ATG6 increases the protein levels of NPR1 and promotes the formation of SINCs-like condensates, possibly caused by ATG6 increasing SA levels in vivo.

## ATG6 maintains the protein stability of NPR1

Maintaining the stability of NPR1 is critical for enhancing plant immunity (*Skelly et al., 2019*). To further verify whether ATG6 regulates NPR1 stability, we co-transfected NPR1-GFP with ATG6-mCherry or mCherry in *N. benthamiana* and performed cell-free degradation assays. Our results showed that NPR1-GFP degradation was significantly delayed when *ATG6* was overexpressed (*Figure 6—figure supplement 1*). A similar trend was observed in *Arabidopsis*, where the NPR1-GFP protein in *ATG6-mCherry × NPR1-GFP* showed a slower degradation rate compared to *NPR1-GFP* during 0–180 min time period in a cell-free degradation assay (*Figure 6a, b*). Moreover, when *Arabidopsis* seedlings were treated with cycloheximide (CHX) to block protein synthesis, we found that NPR1-GFP in *NPR1-GFP* was degraded after CHX treatment for 3–9 hr and the half-life of NPR1-GFP is ~3 hr, while the half-life of NPR1-GFP in *ATG6-mCherry × NPR1-GFP* is ~9 hr (*Figure 6c, d*). In addition, we also analyzed the degradation of NPR1-GFP in *NPR1-GFP* and *NPR1-GFP/atg5* following 100 μM CHX treatment. The results show that the degradation rate of NPR1-GFP in *NPR1-GFP/atg5* plants was similarly to that in *NPR1-GFP* plants (*Figure 6e, f*). These results indicate that ATG6 plays a role in maintaining the stability of NPR1, which may also be related to the fact that ATG6 promotes an increase in free SA in vivo, since SA has the function of increasing NPR1 stability (*Ding et al., 2016*; *Skelly et al., 2019*).

## ATG6 and NPR1 cooperatively inhibit infection of *Pst* DC3000/*avrRps4*

The mRNA expression levels of *ATG6* in Col were significantly increased after 6, 12, and 24 hr under *Pst* DC3000/*avrRps4* treatment (*Figure 7a*). Similarly, both the *ATG6* gene and protein were significantly upregulated under 0.5 mM SA treatment (*Figure 7b, c*). These results suggest that the expression of *ATG6* could be induced by *Pst* DC3000/*avrRps4* and 0.5 mM SA treatment.

Considering that ATG6 increases NPR1 protein levels (*Figure 5a, b*) and promotes its nuclear accumulation (*Figure 3*), as well as maintains NPR1 stability (*Figure 6*), then we studied the role of ATG6–NPR1 interactions in plant immunity. However, studying the function of ATG6 is challenging due to the lethality of homozygous *atg6* mutant (*Qin et al., 2007*; *Harrison-Lowe and Olsen, 2008*; *Patel and Dinesh-Kumar, 2008*). According to our previous report (*Lei et al., 2020*; *Zhang et al., 2023*), *ATG6* was silenced using artificial miRNA$^{ATG6}$ (amiRNA$^{ATG6}$) delivered by the gold nanoparticles (AuNPs). First, the effect of *ATG6* silencing in Col on the plant immune response was investigated. Similar to *atg5*, Col/silencing *ATG6* exhibited more active growth of *Pst* DC3000/*avrRps4* than Col/negative control (NC) after *Pst* DC3000/*avrRps4* infiltration for 3 days (*Figure 7d*). Furthermore, according to the previously reported methods (*Ohira et al., 2017*; *Gomez et al., 2022*), we generated two amiRNA$^{ATG6}$ lines (amiRNA$^{ATG6}$ # 1 and amiRNA$^{ATG6}$ # 2) designed against *ATG6* and placed under the control of a β-estradiol inducible promoter. There were no significant phenotypic differences in amiRNA$^{ATG6}$ # 1 compared to the Col, while amiRNA$^{ATG6}$ # 2 exhibited a slight leaf developmental

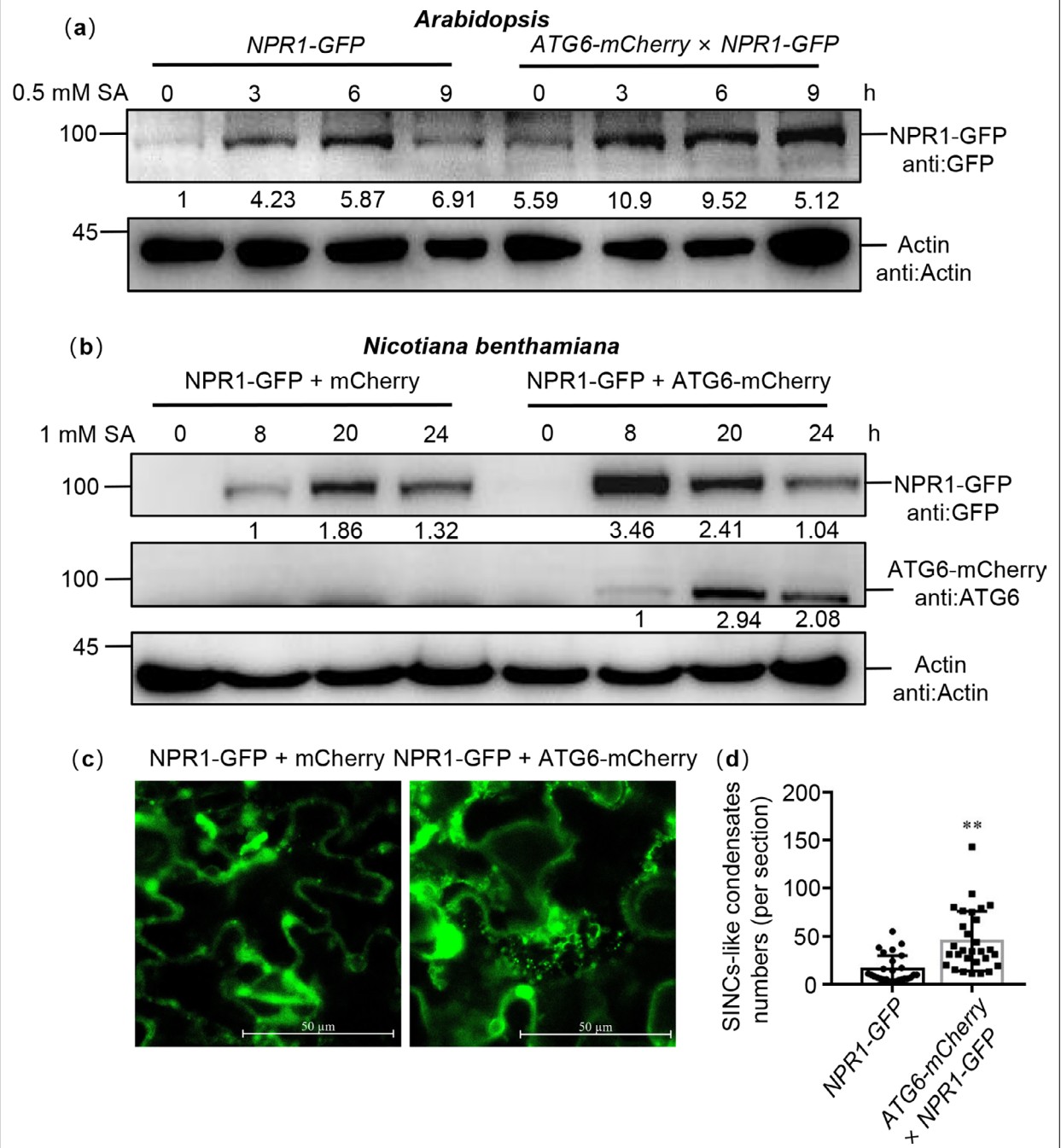

**Figure 5.** ATG6 increases the NPR1 protein levels and the formation of SINCs-like condensates. (**a**) NPR1-GFP protein levels in 7-day-old seedlings of *NPR1-GFP* and *ATG6-mCherry × NPR1-GFP* after 0.5 mM SA treatment for 0, 3, 6, and 9 hr. Numerical values indicate quantitative analysis of NPR1-GFP protein using ImageJ. (**b**) NPR1-GFP protein levels in *N. benthamiana*. ATG6-mCherry + NPR1-GFP, NPR1-GFP + mCherry were co-expressed in *N. benthamiana*. After 2 days, leaves were treated with 1 mM SA for 8, 20, and 24 hr. Total proteins were extracted and analyzed. Numerical values indicate quantitative analysis of NPR1-GFP protein using ImageJ. (**c**) ATG6 promotes the formation of SINCs-like condensates. ATG6-mCherry + NPR1-GFP, NPR1-GFP + mCherry were co-expressed in *N. benthamiana*. After 2 days, leaves were treated with 1 mM SA for 24 hr. Confocal images obtained at excitation with wavelengths of 488 nm, scale bar = 50 μm. (**d**) SINCs-like condensates numbers of per section in (**c**), $n > 10$ sections. ** indicates that the significant difference compared to the control is at the level of 0.01 (Student's *t*-test p value, **$p < 0.01$). All experiments were performed with three biological replicates.

The online version of this article includes the following source data and figure supplement(s) for figure 5:

**Source data 1.** Original files for western blot analysis displayed in *Figure 5a, b*.

*Figure 5 continued on next page*

*Figure 5 continued*

**Source data 2.** PDF file containing original western blots for *Figure 5a, b*, indicating the relevant bands and treatments.

**Source data 3.** Numerical source data files for *Figure 5d*.

**Figure supplement 1.** The protein level of NPR1-GFP in *NPR1-GFP*/silencing *ATG6* and *NPR1-GFP*/Negative control.

**Figure supplement 1—source data 1.** Original files for western blot analysis displayed in *Figure 5—figure supplement 1*.

**Figure supplement 1—source data 2.** PDF file containing original western blots for *Figure 5—figure supplement 1*, indicating the relevant bands and treatments.

**Figure supplement 2.** Expression of *NPR1* in Col and *ATG6-mCherry* under normal and *Pst* DC3000/*avrRps4* treatment.

**Figure supplement 2—source data 1.** Numerical source data files for *Figure 5—figure supplement 2*.

**Figure supplement 3.** Partial co-localization of ATG6-mCherry and SINCs-like condensates.

defect (*Figure 7e, f*). Subsequently, we investigated the expression of *ATG6* following treatment with 100 µM β-estradiol. Our results showed that, after 100 µM β-estradiol treatment for 1–3 days, the expression of *ATG6* in both amiRNA$^{ATG6}$ # 1 and amiRNA$^{ATG6}$ # 2 lines was significantly lower than that in Col. Specifically, the expression of *ATG6* in the amiRNA$^{ATG6}$ #1 and amiRNA$^{ATG6}$ #2 lines decreased by 50–70% compared with Col (*Figure 7g* and *Figure 7—figure supplement 1b*). Furthermore, to assess the function of ATG6 in plant immune, we performed infiltrations of *Pst* DC3000/*avrRps4* after 100 µM β-estradiol treatment for 24 hr. We compared the growth of *Pst* DC3000/*avrRps4* in the amiRNA$^{ATG6}$ lines and Col. The results clearly demonstrate that the growth of *Pst* DC3000/*avrRps4* in amiRNA$^{ATG6}$ # 1 and amiRNA$^{ATG6}$ # 2 was significantly more compared to Col (*Figure 7h*). Moreover, we silenced *ATG6* in *NPR1-GFP* (*NPR1-GFP*/silencing *ATG6*), and *NPR1-GFP*/*atg5* (crossed *NPR1-GFP* with *atg5* to obtain *NPR1-GFP*/*atg5*) was used as an autophagy-deficient control. There was more *Pst* DC3000/*avrRps4* growth in *NPR1-GFP*/silencing *ATG6* and *NPR1-GFP*/*atg5* compared to *NPR1-GFP*/NC after *Pst* DC3000/*avrRps4* infiltration (*Figure 7i*). In contrast, the growth of *Pst* DC3000/*avrRps4* in *NPR1-GFP*, *ATG6-mCherry*, *ATG6-mCherry* × *NPR1-GFP* was significantly lower than that in Col and *npr1* (*Figure 7j*) and was the lowest in *ATG6-mCherry* × *NPR1-GFP* (*Figure 7j*).

These results confirm that ATG6 and NPR1 cooperatively enhance *Arabidopsis* resistance to inhibit *Pst* DC3000/*avrRps4* infection. Together, these results suggest that ATG6 improves plant resistance to pathogens by regulating NPR1.

## Discussion

Although SA signaling and autophagy are related to the plant immune system (*Yoshimoto et al., 2009*; *Munch et al., 2014*; *Wang et al., 2016*), the connection of these two processes in plant immune processes and their interaction is rarely reported. Previous studies have shown that unrestricted pathogen-induced PCD requires SA signaling in autophagy-deficient mutants. SA and its analogue benzo (1,2,3) thiadiazole-7-carbothioic acid (BTH) induce autophagosome production (*Yoshimoto et al., 2009*). Moreover, autophagy has been shown to negatively regulates *Pst* DC3000/*avrRpm1*-induced PCD via the SA receptor NPR1 (*Yoshimoto et al., 2009*), implying that autophagy regulates SA signaling through a negative feedback loop to limit immune-related PCD. Here, we demonstrated

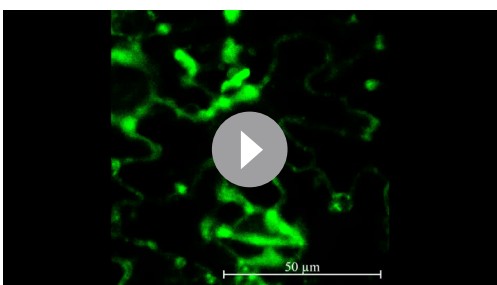

**Video 1.** Localization of NPR1-GFP in *N. benthamiana* co-expressed NPR1-GFP and mCherry.
https://elifesciences.org/articles/97206/figures#video1

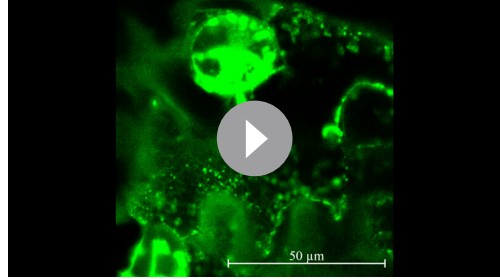

**Video 2.** Localization of NPR1-GFP in *N. benthamiana* co-expressed NPR1-GFP and ATG6-mCherry.
https://elifesciences.org/articles/97206/figures#video2

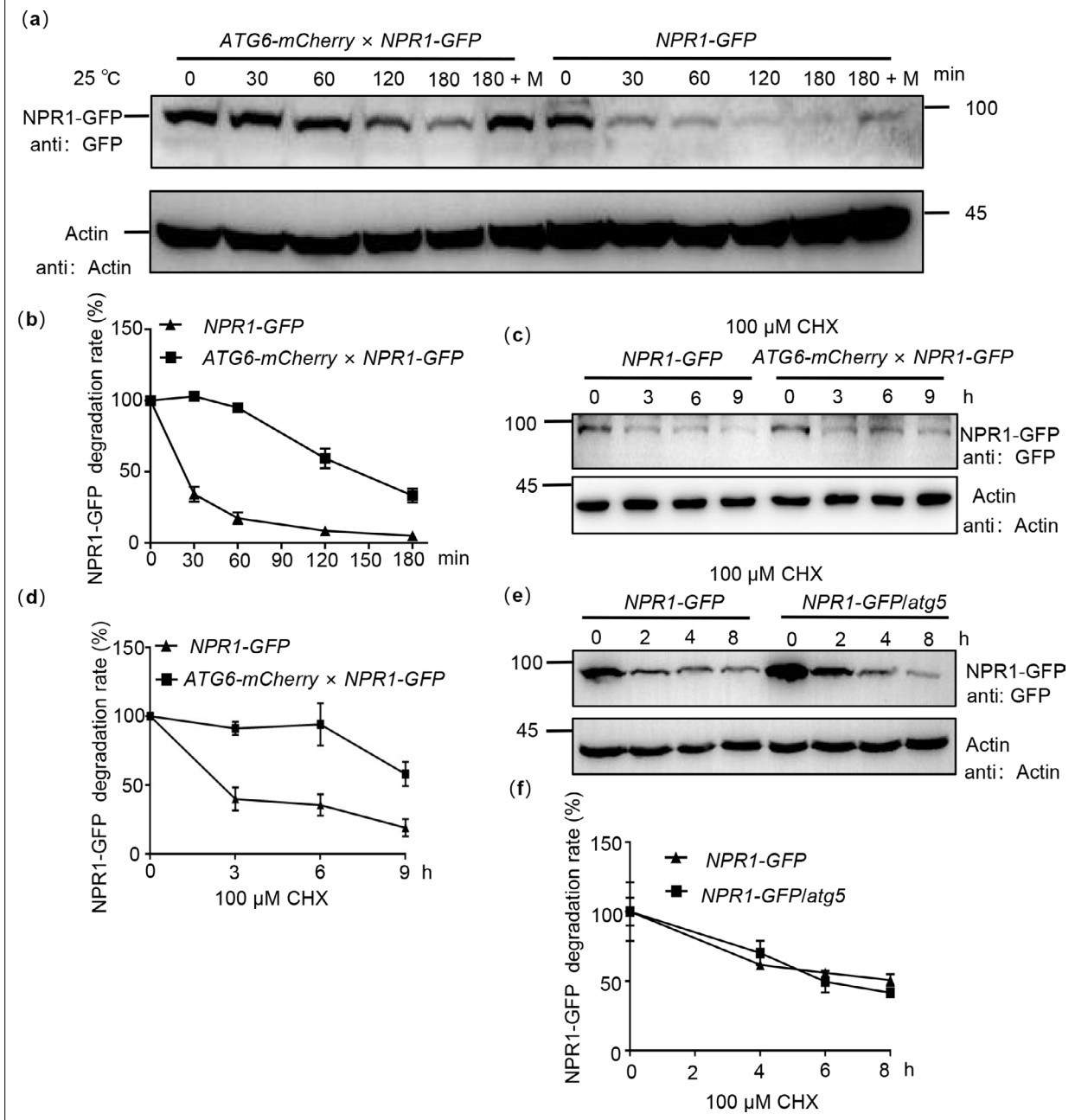

**Figure 6.** ATG6 improves the protein stability of NPR1. (**a**) NPR1-GFP degradation assay in *Arabidopsis*. Total proteins from 7-day-old seedlings of *NPR1-GFP* and *ATG6-mCherry × NPR1-GFP* were extracted, using Actin as an internal reference. 'M' indicates 100 µM MG115 treatment. (**b**) Quantification of NPR1-GFP degradation rates in (**a**) using ImageJ. In (**a, b**), the extracts were incubated for 0–180 min at room temperature (25°C), the degradation rate of NPR1-GFP was analyzed. (**c**) NPR1-GFP protein turnover. Seven-day-old *NPR1-GFP* and *ATG6-mCherry × NPR1-GFP* seedlings were treated with 100 µM cycloheximide (CHX) for different times. Total proteins were analyzed, actin was used as an internal reference. (**d**) Quantification of NPR1-GFP protein turnover rates in (**c**) using ImageJ. (**e**) NPR1-GFP protein turnover. Seven-day-old *NPR1-GFP* and *NPR1-GFP/atg5* seedlings were treated with 100 µM CHX for different times. Total proteins were analyzed, actin was used as an internal reference. (**f**) Quantification of protein levels of NPR1-GFP in (**e**) using ImageJ. All experiments were performed with three biological replicates.

The online version of this article includes the following source data and figure supplement(s) for figure 6:

**Source data 1.** Original files for western blot analysis displayed in *Figure 6a, c, e*.

**Source data 2.** PDF file containing original western blots for *Figure 6a, c, e*, indicating the relevant bands and treatments.

**Source data 3.** Numerical source data files for *Figure 6b, d, f*.

**Figure supplement 1.** ATG6 improves the protein stability of NPR1 in *N. benthamiana*.

*Figure 6 continued on next page*

*Figure 6 continued*

**Figure supplement 1—source data 1.** Original files for western blot analysis displayed in *Figure 6—figure supplement 1a*.

**Figure supplement 1—source data 2.** PDF file containing original western blots for *Figure 6—figure supplement 1a*, indicating the relevant bands and treatments.

**Figure supplement 1—source data 3.** Numerical source data files for *Figure 6—figure supplement 1b*.

**Figure supplement 2.** NPR1-GFP degradation assay in *ATG6-mCherry × NPR1-GFP Arabidopsis*.

**Figure supplement 2—source data 1.** Original files for western blot analysis displayed in *Figure 6—figure supplement 2*.

**Figure supplement 2—source data 2.** PDF file containing original western blots for *Figure 6—figure supplement 2*, indicating the relevant bands and treatments.

that ATG6 increases NPR1 protein levels and nuclear accumulation (*Figures 3 and 5*). Additionally, ATG6 also maintains the stability of NPR1 and promotes the formation of SINCs-like condensates (*Figures 5 and 6*). These findings introduce a novel perspective on the positive regulation of NPR1 by ATG6, highlighting their synergistic role in enhancing plant resistance.

Our results confirmed that *ATG6* overexpression significantly increased nuclear accumulation of NPR1 (*Figure 3*). ATG6 also increases NPR1 protein levels and improves NPR1 stability (*Figures 5 and 6*). Therefore, we consider that the increased nuclear accumulation of NPR1 in *ATG6-mCherry × NPR1-GFP* plants might result from higher levels and more stable NPR1 rather than the enhanced nuclear translocation of NPR1 facilitated by ATG6. To verify this possibility, we determined the ratio of NPR1-GFP in the nuclear localization versus total NPR1-GFP. Notably, the ratio (nucleus NPR1/total NPR1) in *ATG6-mCherry × NPR1-GFP* was not significantly different from that in *NPR1-GFP*, and there is a similar phenomenon in *N. benthamiana* (*Figure 3c–f*). Further we analyzed whether ATG6 affects NPR1 protein levels and protein stability. Our results show that ATG6 increases NPR1 protein levels under SA treatment and ATG6 maintains the protein stability of NPR1 (*Figures 5 and 6*). These results suggested that the increased nuclear accumulation of NPR1 by ATG6 result from higher levels and more stable NPR1.

NPR1 is an important signaling hub of the plant immune response. Nuclear localization of NPR1 is essential to enhance plant resistance (*Kinkema et al., 2000*; *Chen et al., 2021b*), it interacts with transcription factors such as TGAs in the nucleus to activate expression of downstream target genes (*Chen et al., 2019*; *Chen et al., 2021a*). A recent study showed that nuclear-located ATG8h recognizes C1, a geminivirus nuclear protein, and promotes C1 degradation through autophagy to limit viral infiltration in solanaceous plants (*Li et al., 2020*). Here, we confirmed that ATG6 is also distributed in the nucleus and ATG6 is co-localized with NPR1 (*Figures 1d and 2*), suggesting that ATG6 interact with NPR1 in the nucleus. ATG6 synergistically inhibits the infection of *Pst* DC3000/*avrRps4* with NPR1. Chen et al. found that in the nucleus, NPR1 can recruit enhanced disease susceptibility 1 (EDS1), a transcriptional coactivator, to synergistically activate expression of downstream target genes (*Chen et al., 2021a*). Previous studies have shown that acidic activation domains (AADs) in transcriptional activators (such as Gal4, Gcn4, and VP16) play important roles in activating downstream target genes. Acidic amino acids and hydrophobic residues are the key structural elements of AAD (*Pennica et al., 1984*; *Cress and Triezenberg, 1991*; *Van Hoy et al., 1993*). Chen et al. found that EDS1 contains two ADD domains and confirmed that EDS1 is a transcriptional activator with AAD (*Chen et al., 2021a*). Here, we also have similar results that *ATG6* overexpression significantly enhanced the expression of *PR1* and *PR5* (*Figure 4b, c* and *Figure 4—figure supplement 2*), and that the ADD domain containing acidic and hydrophobic amino acids is also found in ATG6 (148–295 AA) (*Figure 4—figure supplement 3*). We speculate that ATG6 might act as a transcriptional coactivator to activate *PRs* expression synergistically with NPR1.

A recent study showed that SA not only enhances plant resistance by increasing NPR1 nuclear import and transcriptional activity, but also promotes cell survival by coordinating the distribution of NPR1 in the nucleus and cytoplasm (*Zavaliev et al., 2020*). Notably, NPR1 accumulated in the cytoplasm recruits other immunomodulators (such as EDS1 and PAD4) to form SINCs to promote cell survival (*Zavaliev et al., 2020*). Similarly, we also found that NPR1 accumulated abundantly in the cytoplasm after SA treatment and that ATG6 significantly increased NPR1 protein levels (*Figures 3c, e and 5a, b*). Obviously, the accumulation of NPR1 in the cytoplasm may be related to ATG6 synergizing with NPR1 to enhance plant resistance. Interestingly, *ATG6* overexpression significantly increased the

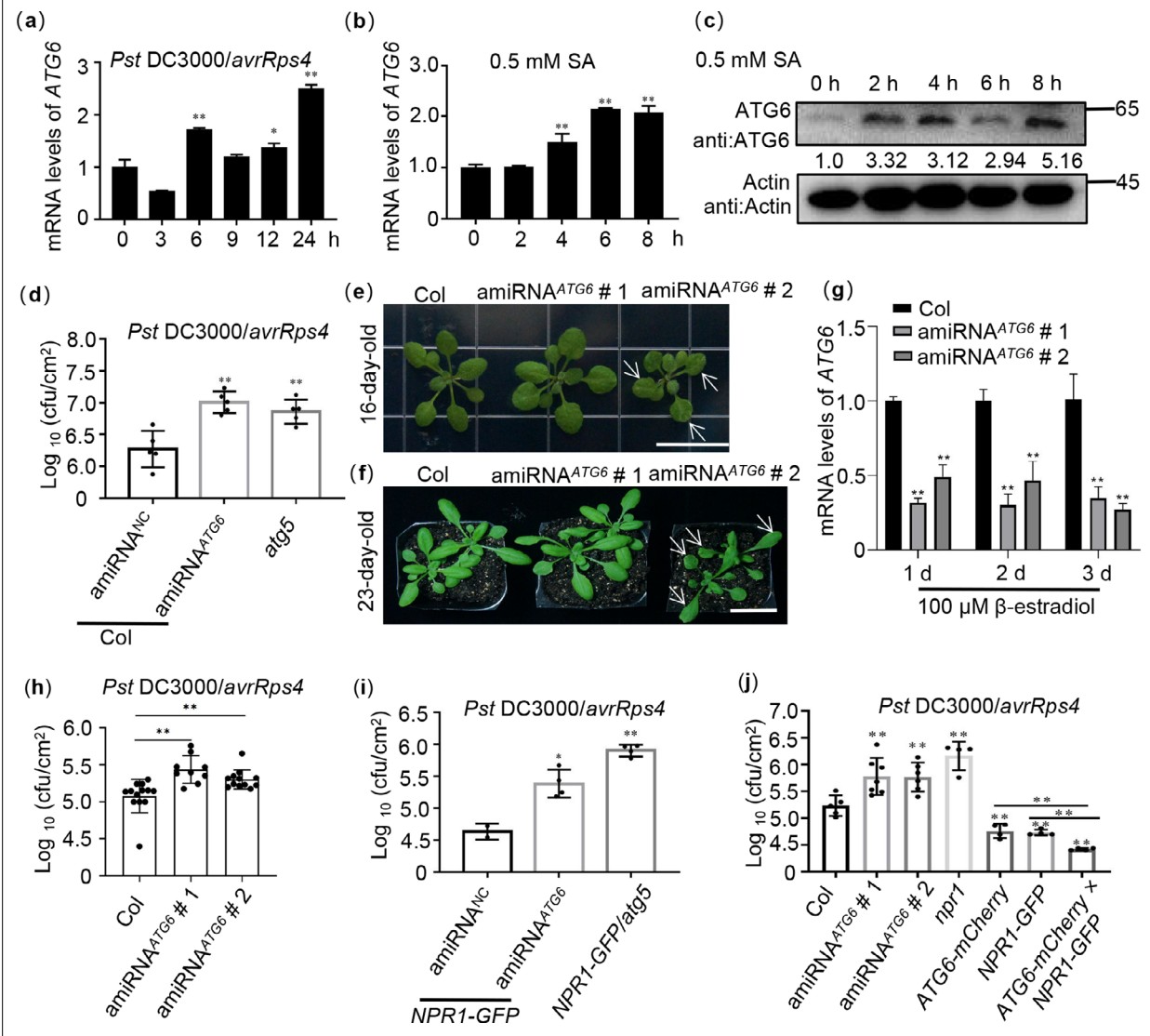

**Figure 7.** ATG6 and NPR1 cooperatively inhibit the growth of *Pst* DC3000/*avrRps4*. (**a**) Expression of *ATG6* under *Pst* DC3000/*avrRps4* infiltration in 3-week-old Col leaves, values are means ± SD (*n* = 3 biological replicates). The AtActin gene was used as the internal control. (**b**) Expression of ATG6 in the presence of 0.5 mM SA in 3-week-old Col leaves, values are means ± SD (*n* = 3 biological replicates). The AtActin gene was used as the internal control. (**c**) The protein levels of ATG6 after 0.5 mM SA in 3-week-old Col leaves. Total leaf proteins from *Arabidopsis* were analyzed, actin was used as an internal reference. Numerical values indicate quantitative analysis of ATG6 protein using ImageJ. (**d**) Growth of *Pst* DC3000/*avrRps4* in Col/ silencing *ATG6* and Col/negative control (NC). (**e**) Phenotypes of 16-day-old amiRNA[ATG6] # 1 and amiRNA[ATG6] # 2. Bar, 1 cm. (**f**) Phenotypes of 23-day-old amiRNA[ATG6] # 1 and amiRNA[ATG6] # 2. Bar, 3 cm. (**g**) Expression of ATG6 in Col, amiRNA[ATG6] # 1 and amiRNA[ATG6] # 2 under infiltration treatment of 100 μM β-estradiol, values are means ± SD (*n* = 3 biological replicates). The AtActin gene was used as the internal control. (**h**) Growth of *Pst* DC3000/*avrRps4* in *Arabidopsis* leaves of amiRNA[ATG6] # 1,  amiRNA[ATG6] # 2 and Col. (**i**) Growth of *Pst* DC3000/*avrRps4* in NPR1 GFP/silencing *ATG6* and NPR1-GFP/NC. (**j**) Growth of *Pst* DC3000/*avrRps4* in *Arabidopsis* leaves of Col, amiRNA[ATG6] # 1,  amiRNA[ATG6] # 2, *npr1*, *NPR1-GFP*, *ATG6-mCherry*, and *ATG6-mCherry* × *NPR1-GFP*. In (**d, h–j**), a low dose of *Pst* DC3000/*avrRps4* (OD$_{600}$ = 0.001) was infiltrated. After 3 days, the growth of *Pst* DC3000/*avrRps4* was counted. * or ** indicates that the significant difference compared to the control is at the level of 0.05 or 0.01 (Student's *t*-test p value, *p < 0.05 or **p < 0.01). All experiments were performed with three biological replicates.

The online version of this article includes the following source data and figure supplement(s) for figure 7:

**Figure supplement 1.** Verification of ATG6 antibody specificity.

**Figure supplement 2.** ATG6 and NPR1 cooperatively inhibit *Pst* DC3000/*avrRps4*-induced cell dead.

**Source data 1.** Original files for western blot analysis displayed in *Figure 7c*.

**Source data 2.** PDF file containing original western blots for *Figure 7c*, indicating the relevant bands and treatments.

**Source data 3.** Numerical source data files for *Figure 7a, b, d, g, h, i, j*.

*Figure 7 continued on next page*

*Figure 7 continued*

**Figure supplement 1—source data 1.** Original files for western blot analysis displayed in *Figure 7—figure supplement 1*.

**Figure supplement 1—source data 2.** PDF file containing original western blots for *Figure 7—figure supplement 1*, indicating the relevant bands and treatments.

**Figure supplement 2—source data 1.** Numerical source data files for *Figure 7—figure supplement 2b*.

formation of SINCs-like condensates (*Figure 5c, d*, *Videos 1 and 2*), which should also be a way for ATG6 and NPR1 to synergistically resist infection of pathogens. We consider that ATG6 promotes the formation of SINCs-like condensates through the dual action of endogenous and exogenous SA. Considering that ATG6 promotes SINCs-like condensates formation, we further examined changes in cell death in Col, amiRNA^{ATG6} # 1, amiRNA^{ATG6} # 2, *npr1*, *NPR1-GFP*, *ATG6-mCherry*, and *ATG6-mCherry × NPR1-GFP* plants. The results of Taipan blue staining showed that *Pst* DC3000/*avrRps4*-induced cell death in *npr1*, amiRNA^{ATG6} # 1, and amiRNA^{ATG6} # 2 was significantly higher compared to Col (*Figure 7—figure supplement 2*). Conversely, *Pst* DC3000/*avrRps4*-induced cell death in *ATG6-mCherry*, *NPR1-GFP*, and *ATG6-mCherry × NPR1-GFP* was significantly lower compared to Col. Notably, *Pst* DC3000/*avrRps4*-induced cell death in *ATG6-mCherry × NPR1-GFP* was significantly lower compared *ATG6-mCherry* and *NPR1-GFP* (*Figure 7—figure supplement 2*). These results suggest that ATG6 and NPR1 cooperatively inhibit *Pst* DC3000/*avrRps4*-induced cell dead.

ATG6 is a common and required subunit of PtdIns3K lipid kinase complexes, which regulates autophagosome nucleation in *Arabidopsis* (*Qi et al., 2017*; *Bozhkov, 2018*). In this study, we also found that ATG6 can maintain the stability of NPR1. Thus, to confirm whether the regulation of NPR1 protein stability by ATG6 is autophagy dependent, we used autophagy inhibitors (Concanamycin A, ConA and Wortmannin, WM) to detect the degradation of NPR1-GFP. Cell-free degradation assays showed that 100 μM MG115 treatment significantly inhibited the degradation of NPR1-GFP. However, 5 μM concanamycin A treatment did not significantly delay NPR1 degradation (*Figure 6—figure supplement 2*). Remarkably, treatment with 30 μM Wortmannin resulted in a slight acceleration of NPR1 degradation, while the combined treatment of ConA and WM significantly expedited the degradation of NPR1 (*Figure 6—figure supplement 2*). This may be related to crosstalk between autophagy and 26S

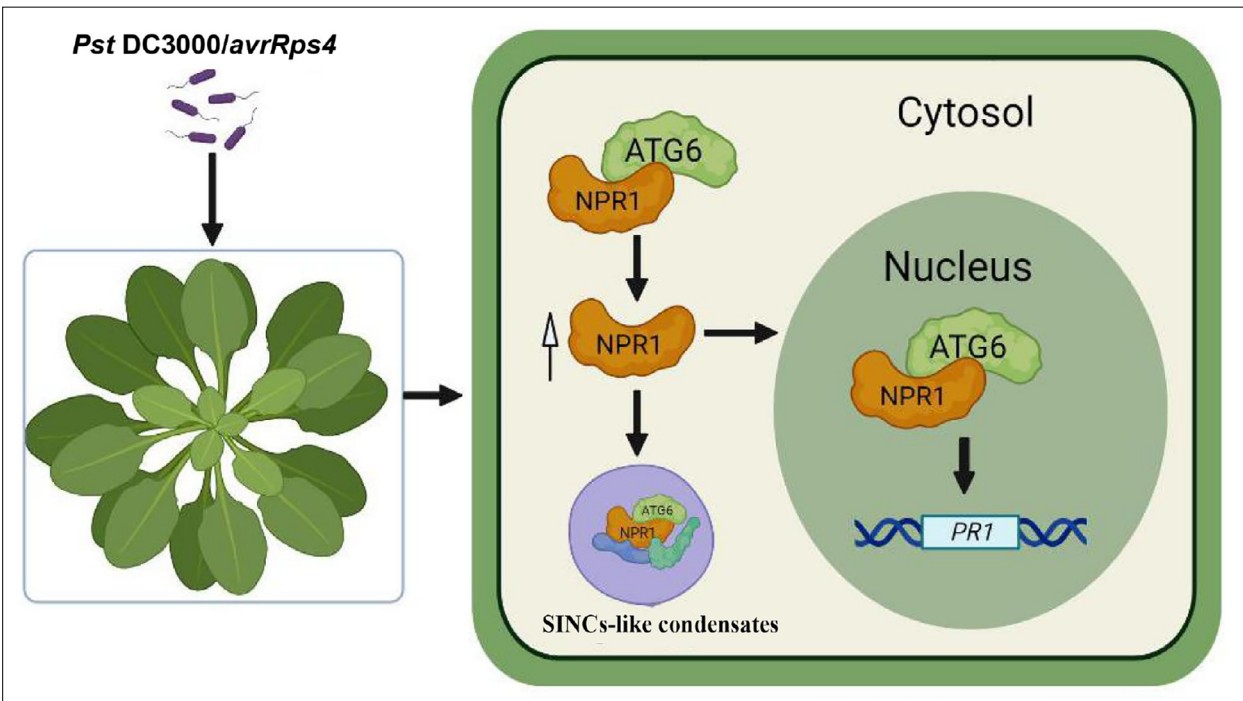

**Figure 8.** Working model for NPR1 regulation by ATG6. ATG6 interacts directly with NPR1 to increase NPR1 protein level and stability, thereby promoting the formation of SINCs-like condensates and increasing the nuclear accumulation of NPR1. ATG6 synergistically activates *PRs* expression with NPR1 to cooperatively enhance resistance to inhibit *Pst* DC3000/*avrRps4* infection in *Arabidopsis*.

Proteasome. It has been demonstrated that autophagy directly regulates the activity of the 26S proteasome under normal conditions or treatment with *Pst* DC3000 (*Marshall et al., 2015*; *Üstün et al., 2018*). Marshall et al. found that the 26S proteasome subunits (RPN1, RPN3, RPN5, RPN10, PAG1, and PBF1) are significantly enriched in autophagy-deficient mutantsunder normal growth conditions (*Marshall et al., 2015*). Treatment with concanamycin A (ConA), an inhibitor of vacuolar-type ATPase, increased the level of the 20S proteasome subunit PBA1 under treatment with *Pst* DC3000 (*Üstün et al., 2018*). In addition, we also analyzed the degradation of NPR1-GFP in *NPR1-GFP* and *NPR1-GFP/atg5* following 100 µM CHX treatment. The results show that the degradation rate of NPR1-GFP in *NPR1-GFP/atg5* plants was similarly to that in *NPR1-GFP* plants (*Figure 6e, f*). These results suggest that deletion of ATG5 do not affect the protein stability of NPR1.

An increasing number of studies have shown that ATGs differentially affect plant immunity. Deletion of ATGs (ATG5, ATG7, ATG10, etc.) leads to reduced resistance of plants to necrotrophic pathogens (*Lai et al., 2011*; *Lenz et al., 2011*; *Minina et al., 2018*). ATGs can directly interact with other proteins to positively regulate plant immunity. In *N. benthamiana*, ATG8f interacts the effector protein βC1 of the cotton *leaf curl multan virus* and promotes its degradation to limit pathogen infection (*Haxim et al., 2017*). Notably, ATG18a can interact with WRKY33 transcription factor to synergistically against *Botrytis* infection (*Lai et al., 2011*). Our evidence shows that ATG6 interacts with NPR1 and works together to counteract pathogen infection by positively regulating NPR1 and SA levels in vivo. In conclusion, we unveil a novel relationship in which ATG6 positively regulates NPR1 in plant immunity (*Figure 8*). ATG6 interacts with NPR1 to synergistically enhance plant resistance by regulating NPR1 protein levels, stability, nuclear accumulation, and formation of SINCs-like condensates.

# Materials and methods

## Plasmid construction

Details of plasmid construction primer used are listed in *Appendix 2—table 1* and *Appendix 2—table 2*, methods are listed in Appendix 3—method 1, The mapping of vectors is listed in Appendix 4.

## Plant material

### Arabidopsis

*35S::NPR1-GFP* (in *npr1-2* background) and *npr1-1* were kindly provided by Dr. Xinnian Dong of Duke University; *atg5-1* (SALK_020601).

*UBQ10::ATG6-mCherry*, *UBQ10::ATG6-GFP*, and amiRNA[ATG6] lines were obtained by *Agrobacterium* transformation (*Clough and Bent, 1998*). ATG6, NPR1 double overexpression of *Arabidopsis* (*ATG6-mCherry × NPR1-GFP*) and *NPR1-GFP/atg5* were obtained by crossing, respectively.

Full description of the *Arabidopsis* screening is included Appendix 3—method 2. Details of plant material are listed in *Appendix 2—table 3*.

## Growth conditions

### Arabidopsis thaliana

All *Arabidopsis thaliana* (*Arabidopsis*) seeds were treated in 10% sodium hypochlorite for 7 min, washed with ddH$_2$O, and treated in 75% ethanol for 30 s, finally washed three times with ddH$_2$O. Seeds were sown in 1/2 MS with 2% sucrose solid medium, vernalized at 4°C for 2 days.

For 7-day-old *Arabidopsis* seedling cultures, the plates were placed under the following conditions: daily cycle of 16 hr light (~80 µmol m$^{-2}$. s$^{-1}$) and 8 hr dark at 23 ± 2°C.

For 3-week-old *Arabidopsis* cultures, after 7 days of growth on the plates, the seedlings were transferred to soil for further growth for 2 weeks under the same conditions (*Zhang et al., 2018a*).

### N. benthamiana

For 3-week-old *N. benthamiana* cultures, seeds were sown in the soil and vernalized at 4°C for 2 days. After 10 days of growth on soil, the seedlings were transferred to soil for further growth for 2 weeks under the same conditions (*Jiao et al., 2019*).

## Treatment conditions
### Treatment of 7-day-old seedlings
#### For SA treatment
Seven-day-old *Arabidopsis* seedlings were transferred to 1/2 MS liquid medium containing 0.5 mM SA for 0, 3, and 6 hr, respectively. The corresponding results are shown in *Figures 2f, g, 3c, d , and 5a*.

#### For CHX treatment
Seedlings of *Arabidopsis* (7 days) were transferred to 1/2 MS liquid medium containing 100 µM CHX for 0, 3, 6, and 9 hr, respectively. The corresponding results are shown in *Figure 6c, e*.

### Treatment of 3-week-old *Arabidopsis*
#### For silencing ATG6 in Col and NPR1-GFP
As previously described (*Lei et al., 2020*; *Zhang et al., 2022*; *Zhang et al., 2023*), 1 mM gold nanoparticles (AuNPs) were synthesized. The artificial microRNA (amiRNA)$^{ATG6}$ (UCAAUUCUAGGA UAACUGCCC) was designed based on the Web MicroRNA Designer (http://wmd3.weigelworld.org/) platform. The complementary sequence of amiRNA$^{ATG6}$ is located on the eighth exon of the *ATG6* gene. The sequence of 'UUCUCCGAACGUGUCACGUTT' was used as a negative control (NC). NC is a universal negative control without species specificity (*Gao et al., 2018*; *Lei et al., 2020*). amiRNA$^{ATG6}$ and amiRNA$^{NC}$ synthesized by Suzhou GenePharma. AuNPs (1 mM) and amiRNA$^{ATG6}$ (20 µM) were incubated at a 9:1 ratio for 30 min at 25°C, 50 rpm. After incubation, a mixture of AuNPs and amiR-NA$^{ATG6}$ was diluted 15-fold with the infiltration buffer (pH 5.7, 10 mM 2-Morpholinoethanesulphonic acid (MES), 10 mM MgCl$_2$) and infiltrated through the abaxial leaf surface into 3-week-old Col or *NPR1-GFP* for 1–3 days. The third day was chosen as material for *ATG6* silencing. After the third day of AuNPs-amiRNA$^{ATG6}$ and AuNPs-amiRNA$^{NC}$ infiltration, *Pst* DC3000/*avrRps4* was infiltrated, and then growth of *Pst* DC3000/*avrRps4* was detected.

#### For β-estradiol treatment
100 µM β-estradiol was infiltrated to treat 3-week-old *Arabidopsis* leaves. After 24 hr of treatment with β-estradiol, *Pst* DC3000/*avrRps4* was infiltrated and then growth of *Pst* DC3000/*avrRps4* was detected after 3 days.

#### For *Pst* DC3000/*avrRps4* infiltration
Infiltration with *Pst* DC3000/*avrRps4* was performed as previously described (*Wang et al., 2016*; *Skelly et al., 2019*). Full description of the *Pst* DC3000/*avrRps4* culture is included in Appendix 3— method 3.

#### For SA treatment
For 3-week-old Col, 0.5 mM SA was infiltrated into the leaves for 0, 2, 4, 6, and 8 hr. The corresponding results are shown in *Figure 7b, c*.

## Y2H assay
Y2H experiments were performed according to the previously described protocol (*Fu et al., 2012*). Full description of Y2H is included in Appendix 3—method 4.

## Pull-down assays in vitro
500 µl of GST, GST-ATG6, and SnRK2.8-GST were incubated with GST-tag Purification Resin (Beyotime, P2250) for 2 hr at 4°C. The mixture was then centrifuged at 1500 × *g* for 1 min at 4°C, and the resin was washed three times with PBS buffer. Next, the GST-tag purification resin was incubated with the NPR1-His for 2 hr at 4°C. After washing three times with PBS buffer, 2× sample buffers were added to the resin and denatured at 100°C for 10 min. The resulting samples were then used for western blotting analysis. Full description of prokaryotic proteins expression is included in Appendix 3–method 5.

## Co-immunoprecipitation

0.5 g leaves of *N. benthamiana* transiently transformed with ATG6-mCherry + GFP and ATG6-mCherry + NPR1-GFP were fully ground in liquid nitrogen and homogenized in 500 µl of lysis buffer (50 mM Tris-HCl pH 7.5, 150 mM NaCl, 0.5 mM EDTA, 5% glycerol, 0.2% NP40, 1 mM Phenylmethylsulfonyl fluoride (PMSF), 40 µM MG115, protease inhibitor cocktail 500× and phosphatase inhibitor cocktail 5000×). The samples were then incubated on ice for 30 min, and centrifuged at 10,142 × *g* (TGL16, Cence, Hunan, China) for 15 min at 4°C. The supernatant (500 µl) was incubated with 20 µl of GFP-Trap Magnetic Agarose beads (ChromoTek, gtma-20) in a 1.5-ml Eppendorf tube for 2 hr by rotating at 4°C. After incubation, the GFP-Trap magnetic Agarose beads were washed three times with cold wash buffer (50 mM Tris-HCl pH 7.5, 150 mM NaCl, 0.5 mM EDTA) and denatured at 75°C for 10 min after adding 2× sample buffer. Western blotting was performed with antibodies to ATG6 and GFP.

## Nuclear and cytoplasmic separation

Nuclear and cytoplasmic separation were performed according to the previously described method (*Kinkema et al., 2000*). Full description of nuclear and cytoplasmic separation is given in Appendix 3—method 6.

## Protein degradation in vitro

Protein degradation assays were performed according to a previously described method (*Spoel et al., 2009*; *Saleh et al., 2015*). Full description of protein degradation is included in Appendix 3—method 7.

## Protein extraction and western blotting analysis

Protein extraction and western blotting were performed as previously described (*Lei et al., 2020*; *Zhang et al., 2022*). Protein was denatured at 100°C for 10 min. NPR1 protein was denatured at 75°C for 10 min (*Lei et al., 2020*). Full description is included in Appendix 3—method 8. Antibody information is presented in *Appendix 2—table 4*.

## Confocal microscope observation

### For nuclear localization of NPR1-GFP observation

Seven-day-old seedlings of *NPR1-GFP* and *ATG6-mCherry × NPR1-GFP* were sprayed with 0.5 mM SA for 0 and 3 hr. GFP and mCherry fluorescence signals in leaves were observed under the confocal microscope (Zeiss LSM880). Statistical data were obtained from three independent experiments, each comprising five individual images, resulting in a total of 15 images analyzed for this comparison.

### For the BiFC assay

*Agrobacterium* was infiltrated into *N. benthamiana* as previously described (*Jiao et al., 2019*). Fluorescence signals were observed after 3 days. The full description of BiFC is contained in Appendix 3—methods 9 and 10.

### For the observation of SINCs-like condensates

*Agrobacterium* was infiltrated into *N. benthamiana*. After 2 days, the leaves were treated in 1 mM SA solution for 24 hr, and then fluorescence signals were observed. At least 20 image sets were obtained and analyzed. A full description of SINCs-like condensates observation is included in Appendix 3—method 11.

### For growth of *Pst* DC3000/*avrRps4*

A low dose (OD$_{600}$ = 0.001) of *Pst* DC3000/*avrRps4* was used for the infiltration experiments. After 3 days, the colony count was counted according to a previous description (*Wang et al., 2016*; *Lei et al., 2020*). Full description of the growth of *Pst* DC3000/*avrRps4* is given in Appendix 3—method 12.

## Free SA measurement

Free SA was extracted from 3-week-old *Arabidopsis* using a previously described method (*Wang et al., 2016*; *Gong et al., 2020*). Free SA was measured by high-performance liquid chromatography (Shimadzu LC-6A, Japan). Detection conditions: 294 nm excitation wavelength, 426 nm emission wavelength.

## Real-time quantitative PCR

Total RNA was extracted from *Arabidopsis* (100 mg) using Trizol RNA reagent (Invitrogen, 10296-028, Waltham, MA, USA). Real-time quantitative PCR (RT-qPCR) assays were performed as previously described (*Zhang et al., 2018a*; *Zhang et al., 2022*). All primers for RT-qPCR are listed individually in *Appendix 2—table 5*. Full description of RT-qPCR is included in Appendix 3—method 13.

## Trypan blue staining

The leaves of 3-week-old Col, amiRNA$^{ATG6}$ # 1, amiRNA$^{ATG6}$ # 2, *npr1*, *NPR1-GFP*, *ATG6-mCherry*, and *ATG6-mCherry* × *NPR1-GFP* plants, located in the fifth and sixth positions, were infiltrated with *Pst* DC3000/*avrRps4*. After 3 days, the leaves were excised and subjected to a 1-min boiling step in trypan blue staining buffer (consisting of 10 g phenol, 10 ml glycerol, 10 ml lactic acid, 10 ml ddH$_2$O, and 10 mg trypan blue), followed by destaining three times at 37°C in 2.5 mg/ml chloral hydrate.

## Statistical analysis

All quantitative data in this study were presented as mean ± SD. The experimental data were analyzed by a two-tailed Student's *t*-test. Significance was assigned at p values <0.05 or <0.01.

## Acknowledgements

We thank Dr. Xinnian Dong (Duke University, USA), Dr. ZhengQing Fu (University of South Carolina), and Dr. Sheng Li (South China Normal University) for their help and contribution. This research was supported by the National Natural Science Foundation of China [Grant Number 31570256].

## Additional information

### Funding

| Funder | Grant reference number | Author |
| --- | --- | --- |
| National Natural Science Foundation of China | 31570256 | Wenli Chen |

The funders had no role in study design, data collection and interpretation, or the decision to submit the work for publication.

### Author contributions

Baihong Zhang, Data curation, Software, Formal analysis, Validation, Investigation, Visualization, Writing – original draft, Writing – review and editing; Shuqin Huang, Data curation, Validation, Investigation; Shuyu Guo, Validation, Visualization; Yixuan Meng, Yuzhen Tian, Yue Zhou, Hang Chen, Xue Li, Validation, Investigation; Jun Zhou, Supervision, Funding acquisition, Project administration; Wenli Chen, Supervision, Funding acquisition, Writing – original draft, Project administration, Writing – review and editing

### Author ORCIDs

Jun Zhou http://orcid.org/0000-0001-9655-6588
Wenli Chen https://orcid.org/0000-0002-4768-3844

Reviewer #1 (Public Review): https://doi.org/10.7554/eLife.97206.5.sa1
Reviewer #2 (Public Review): https://doi.org/10.7554/eLife.97206.5.sa2
Author response https://doi.org/10.7554/eLife.97206.5.sa3

# Additional files

## Supplementary files
MDAR checklist

## Data availability
All data generated or analyzed during this study are included in the manuscript and supporting files.

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

## Appendix 1

### Result 1: NPR1 and its paralogs NPR3/NPR4 physically interact with multiple ATGs

NPR1 and its paralogs NPR3/NPR4, which frequently interact with other proteins to regulate plant immune responses (*Backer et al., 2019*; *Chen et al., 2019*). To identify ATGs that interact with NPRs, we performed yeast two-hybrid (Y2H) screens using NPRs as bait. Interestingly, ATG6 interacted with NPR1, NPR3, and NPR4, respectively, and different concentrations of SA treatment did not significantly affect their interaction (*Figure 1—figure supplement 1a*). ATG8e interacted with NPR3 and NPR4, respectively, and there was no significant effect of different concentrations of SA treatment on their interactions (*Figure 1—figure supplement 1b*). NPR1 interacted with ATG8e and ATG8d, respectively, and their interactions were inhibited with increasing SA content (*Figure 1—figure supplement 1b, e*). NPR3 did not interact with ATG8f, ATG8d, and ATG8g under normal conditions, but SA significantly promoted their interactions (*Figure 1—figure supplement 1c–f*). ATG8g had a weak interaction with NPR4 under normal conditions, and SA significantly promoted their interactions (*Figure 1—figure supplement 1c*). NPR1 did not interact with ATG8g and ATG8f under normal and SA treatment (*Figure 1—figure supplement 1c, d*). NPR4 did not interact with ATG8f under normal and SA treatment (*Figure 1—figure supplement 1d*). NPR1 is an important positive regulator of the plant immune response (*Chen et al., 2021b*). So far, nine ATG8 isoforms have been identified in *Arabidopsis*, and considering the possible redundancy between ATG8 protein (*Bu et al., 2020*), we mainly further investigated the function of ATG6 interactions with NPR1 in plant immune response.

### Result 2: Overexpression of *ATG6* delays carbon starvation-induced leaf senescence, and *ATG6-GFP* and *ATG6-mCherry* fusion proteins are functional

AtATG6 is a member of the class III phosphatidylinositol 3-kinase family (PtdIns3K), which regulates autophagosome nucleation in *Arabidopsis* (*Qi et al., 2017*; *Bozhkov, 2018*). Previous studies have shown that one of the most prominent features of autophagy-deficient mutants is hypersensitivity to carbon starvation with premature senescence, and shorter growth cycles (*Yoshimoto et al., 2009*; *Bozhkov, 2018*; *Huang et al., 2019*). In contrast, activated autophagy delays carbon starvation-induced leaf senescence (*Yoshimoto et al., 2009*; *Bozhkov, 2018*; *Huang et al., 2019*). To verify whether the ATG6-GFP and ATG6-mCherry fusion proteins are functional in *Arabidopsis*. We analyzed phenotypic changes of Col, amiRNA^ATG6 # 1, amiRNA^ATG6 # 2, *ATG6-GFP*, and *ATG6-mCherry* under carbon starvation. Autophagy-deficient mutant *atg5* was used as a positive control for leaf senescence. When the detached rosette leaves from 3-week-old *Arabidopsis* were treated in the dark for 4 days, the leaf phenotypes of *ATG6-GFP* and *ATG6-mCherry* were greener than Col and the chlorophyll content in *ATG6-GFP* and *ATG6-mCherry* was also significantly higher than Col (*Figure 3—figure supplement 3*). The severity of senescence followed the order: *atg5* > amiRNA^ATG # 2>amiRNA^ATG # 1 > Col > *ATG6-GFP* or *ATG6-mCherry* (*Figure 3—figure supplement 3*). These results suggest that overexpression of *ATG6* delays carbon starvation-induced leaf senescence, and ATG6-GFP and ATG6-mCherry fusion proteins are functional.

## Appendix 2

**Appendix 2—table 1.** Plasmid in this study.

| Gene name | Vector name | Subcloning method | Source |
|---|---|---|---|
| *ATG6* | pGADT7 | ClonExpress II One Step Cloning Kit | This paper |
| *NPR1* | pGBKT7 | ClonExpress II One Step Cloning Kit | This paper |
| *NPR1-C* | pGBKT7 | ClonExpress II One Step Cloning Kit | This paper |
| *NPR1-N* | pGBKT7 | ClonExpress II One Step Cloning Kit | This paper |
| *SnRK2.8* | pGADT7 | ClonExpress II One Step Cloning Kit | This paper |
| *ATG6* | 35s-gene-cYFP | ClonExpress II One Step Cloning Kit | This paper |
| *NPR1* | 35s-gene-nYFP | ClonExpress II One Step Cloning Kit | This paper |
| *SnRK2.8* | 35s-gene-cYFP | ClonExpress II One Step Cloning Kit | This paper |
| *SnRK2.8* | pGEX-4T-1 | Double digests and T4 DNA ligase | This paper |
| *ATG6* | pGEX-4T-1 | Double digests and T4 DNA ligase | This paper |
| *ATG6* | 1300:UBQ-mCherry | Double digests and T4 DNA ligase | This paper |
| *ATG6* | 1300:UBQ-eGFP | Double digests and T4 DNA ligase | This paper |
| *GFP* | 1300:UBQ-eGFP | N/A | This paper |
| *NPR1* | pET32a | Provided by Dr. ZhengQing Fu of University of South Carolina | |
| *NPR1* amiRNA*ATG6* | pCB302-GFP pERM10M | Provided by Dr. ZhengQing Fu of University of South Carolina Double digests and T4 DNA ligase This paper | |

**Appendix 2—table 2.** Primers for vector construction.

| Vector name | Primers (5′–3′) |
|---|---|
| For Y2H assay | |
| ATG6-AD (pGADT7) | F: GAGGCCAGTGAATTCCACCCGATGAGGAAAGAGGAGATTCCAG<br>R: CCCGTATCGATGCCCACCCCTAAGTTTTTTTACATGAAGGCT |
| NPR1-BD (pGBKT7) | F: CATGGAGGCCGAATTCCCGATGGACACCACCATTGATG<br>R: CAGGTCGACGGATCCCCTCACCGACGACGATGAGAG |
| NPR1-C-BD (pGBKT7) | F: CATGGAGGCCGAATTCCCGCTTCATTTCGCTGTTGCAT |
| NPR1-N-BD (pGBKT7) | R: CAGGTCGACGGATCCCCTCAAGCACACGCATCATCTAGAT |
| SnRK2.8-AD (pGADT7) | F: CATGGAGGCCGAATTCCCGATGGAGAGGTACGAAATAGTGAAG<br>R: CAGGTCGACGGATCCCCTCACAAAGGGGAAAGGAGATCAGCGGT |
| For BiFC assay | |
| ATG6 -cYFP (35s-gene-cYFP) | F: CGACGGTACCGCGGGCCCGGGATGAGGAAAGAGGAGATTCCAG<br>R: CACGCTGCCCAGGATCCCGGGAGTTTTTTTACATGAAGGCT |
| NPR1-nYFP (35s-gene-nYFP) | F: CGACGGTACCGCGGGCCCGGGATGGACACCACCATTGATG<br>R: GCTCACCATCAGGATCCCGGGCCGACGACGATGAGAGAG |
| SnRK2.8 -cYFP (35s-gene-cYFP) | F: CGACGGTACCGCGGGCCCGGGATGGAGAGGTACGAAATAGTGAAG<br>R: CACGCTGCCCAGGATCCCGGGCAAAGGGGAAAGGAGATCAGCGGT |
| For plant transformation | |

*Appendix 2—table 2 Continued on next page*

*Appendix 2—table 2 Continued*

| Vector name | Primers (5′–3′) |
| --- | --- |
| ATG6-mCherry<br>(1300:UBQ-mCherry) ATG6-GFP<br>(1300:UBQ-GFP)<br>amiRNA$^{ATG6}$ I<br>amiRNA$^{ATG6}$ II<br>amiRNA$^{ATG6}$ III<br>amiRNA$^{ATG6}$ IV<br>miRNA 319 F<br>miRNA 319 R | F: cagACTAGTATGAGGAAAGAGGAGATTCCAG<br>R: cagACTAGTAGTTTTTTTTACATGAAGGCTTACTAG<br>F: cagACTAGTATGAGGAAAGAGGAGATTCCAG<br>R: cagACTAGTAGTTTTTTTTACATGAAGGCTTACTAG<br>miR-s: GATCAATTCTAGGATAACTGCCCCTCTCTTTTGTATTCCA<br>miR-a: AGGGGCAGTTATCCTAGAATTGATCAAAGAGAATCAATGA<br>miR*s: AGGGACAGTTATCCTTGAATTGTTCACAGGTCGTGATATG<br>miR*a: GAACAATTCAAGGATAACTGTCCCTACATATATATTCCTA<br>F: CGCGGATCCCAAACACACGCTCGGACGCATATT<br>R: TCCCCCGGGCATGGCGATGCCTTAAATAAAGATAAACCC |
| GST-ATG6 (pGEX-4T-1) | F: GAATTCATGAGGAAAGAGGAGATTCC<br>R: GTCGACAGTTTTTTTACATGAAGGCTTACTAG |
| GST-SnRK2.8 (pGEX-4T-1) | F: CGCGGATCCATGGAGAGGTACGAAATAGTGAAG<br>R: CCGCTCGAGCAAAGGGGAAAGGAGATCAGCGGT |

**Appendix 2—table 3.** Plant materials.

| Name | Source |
| --- | --- |
| *NPR1-GFP/atg5* | This paper (crossing) |
| *ATG6-mCherry×NPR1-GFP/npr1-2* | This paper (crossing) |
| *ATG6-mCherry*<br>*ATG6-GFP* amiRNA$^{ATG6}$ | This paper (floral dip method)<br>This paper (floral dip method)<br>This paper (floral dip method) |
| *atg5-1* | SALK_020601C |
| *NPR1-GFP* (in *npr1-2* background) | Provided by Dr. Xinnian Dong of Duke University |
| *npr1-1* | Provided by Dr. Xinnian Dong of Duke University |

**Appendix 2—table 4.** Antibody information.

| Antibodies | Dilution | Identifier | Source |
| --- | --- | --- | --- |
| anti-GFP | 1:3000 | CAT#A-6455 | Invitrogen |
| anti-GST | 1:5000 | CAT#AT0027 | Engibody |
| anti-His | 1:2000 | CAT#AH367 | Beyotime |
| anti-Actin | 1:3000 | CAT#AT0004 | Engibody |
| anti-H3 | 1:3000 | CAT#NB500-171 | Novus Biologicals |
| anti-ATG6 | 1:200 | Peptide, C-KEKKKIEEEERK | Abmart |

**Appendix 2—table 5.** Primers of RT-qPCR.

| Genes | Primers (5′–3′) |
| --- | --- |
| *AtActin2*<br>*AtNPR1* | F: GGTAACATTGTGCTCAGTGGTGG<br>R: AACGACCTTAATCTTCATGCTGC<br>F:GATCGCAAAACAAGCCACTATGG<br>R:ATCGAGCAGCGTCATCTTCAATT |
| *AtATG6* | F:TCCTCCATACGATGTGTAACTATTTCC<br>R:GCTCATAAGTTTCGTTGTTGCTGT |
| *AtPR1*<br>*AtPR5*<br>*AtICS1* | F:TGTAGCTCTTGTAGGTGCTC<br>R:AACTCCATTGCACGTGTTCG<br>F:AGTTCCTCCCGTCACTCTGG<br>R:TCCTCCGGATGGTCTTATCC<br>F: GAGACTTACGAAGGAAGATGATGAG<br>R:TGATCCCGACTGCAAATTCACTCTC |

## Appendix 3

### Method 1: Plasmid construction

### For plant transformation

The *ATG6* coding regions were prepared by PCR with Ex Taq DNA polymerase (TaKaRa, RR001A, Dalian, China) and cloned into 1300-UBQ-mCherry or 1300-UBQ-GFP via double digests and T4 DNA ligase. The amiRNA^ATG6 recombinant plasmid was constructed by double digestion. The amplification primers for the amiRNA^ATG6 precursor, including miR-s (primer I), miR-a (primer II), miR*s (primer III), and miR*a (primer IV), were designed using the artificial microRNA (amiRNA) design platform Web MicroRNA Designer (WMD3, http://wmd3.weigelworld.org). To generate the stem-loop structure of the amiRNA^ATG6 precursor, pCB302-amiR-GFP was utilized as a template (*Zhang et al., 2018b*). An overlap extension PCR method (*Niu et al., 2006*; *Carbonell et al., 2015*), was employed for the synthesis of amiRNA^ATG6. The specific procedure involved two rounds of amplification. First, miR319 F and primer IV, miR319 R and primer I, primer II, and primer III were used for the first round of amplification. Subsequently, a combination of the first-round PCR amplification products served as templates for the second round of PCR amplification using miR319 F and miR319 R primers. The resulting products from the second-round PCR were digested with appropriate restriction endonucleases (BamH I and Sma I) and then ligated to the 1300-UBQ-mCherry and pERM10M vectors using T4 DNA ligase.

### For Y2H assay

For the interaction of ATG6 and NPR1.The coding regions of *ATG6*, *SnRK2.8*, *NPR1*, *NPR1-N* (1–984 bp), *NPR1-C* (984–1782 bp) were prepared by PCR with Ex Taq DNA polymerase (TaKaRa, RR001A, Dalian, China) using the primers containing 15–20 bp homologous sequence of the linearized pGADT7 or pGBKT7 vector. The *ATG6* and *SnRK2.8* coding regions were cloned into pGADT7 via ClonExpress II One Step Cloning Kit (Vazyme, C112-02, Nanjing, China); the coding regions of *NPR1*, *NPR1-N*, and *NPR1-C* were cloned into pGBKT7, respectively. For interaction of ATGs and NPRs. The coding region of ATGs and NPRs was generated by PCR with Ex Taq DNA polymerase using the primers containing Gateway attB sites. The amplified fragment was cloned into the pDONR207 vector by the BP Clonase II reaction (Invitrogen, 11789-020, Waltham, MA, USA). Each positive clone was inserted into the gateway destination pDEST-GBKT7 and pDEST-GADT7 for yeast transformation by LR Clonase II (Invitrogen, 11791-020, Waltham, MA, USA).

### For pull-down assay

The *ATG6* and *SnRK2.8* coding regions were generated by PCR with Ex Taq DNA polymerase (TaKaRa, RR001A, Dalian, China) and cloned into pGEX-4T-1 via double digests (pGEX-4T-1 was provided by Dr. Sheng Li of South China Normal University) and T4 DNA ligase; NPR1-His (pET32a-NPR1) was provided by Dr. ZhengQing Fu of University of South Carolina.

### For the BiFC assay

The coding regions of *ATG6*, *SnRK2.8*, and *NPR1* were generated by PCR with Ex Taq DNA polymerase (TaKaRa, RR001A, Dalian, China) using the primers containing a homologous sequence (15–20 bp) of linearized 35s-gene-nYFP or 35s-gene-cYFP vector. The *ATG6* and *SnRK2.8* coding regions were cloned into 35s-gene-cYFP via the ClonExpress II One Step Cloning Kit (Vazyme, C112-02, Nanjing, China); NPR1 were cloned into 35s-gene-nYFP.

### Method 2: *Arabidopsis thaliana* screening

For *UBQ10::ATG6-mCherry* and *UBQ10::*ATG6-GFP plants, *Agrobacterium tumefaciens* strain GV3101 harboring ATG6-mCherry and ATG6-GFP was used for Col transformation. *Agrobacterium tumefaciens* strain GV3101 harboring ATG6-mCherry and ATG6-GFP was used for Col transformation. *Agrobacterium tumefaciens* was cultured in LB solid medium containing 25 mg/l rifampicin (rif) and 50 mg/l kanamycin (kana) for 2 days, and then a single clone was grown in LB liquid medium containing 25 mg/l rif and 50 mg/l kana for 16–18 hr at 28°C, 180 rpm. Centrifuge the bacteria at 4000 × *g* for 10 min, then resuspend the bacteria in 100 ml of permeate (5% sucrose, 0.05% Silwet-77, mix well before dipping, OD$_{600}$ = 0.8–1.0). The mossy flowering plants were selected and the pods and pollinated flowers were removed before transformation. The inflorescence of *Arabidopsis* was placed in the transformation medium containing *Agrobacterium tumefaciens* for 1 min. Incubate and moisturize in the dark for 2 days. Cultivation was continued until plants matured

and seeds were collected to screen positive plants. Positive plants with 30 mg/l hygromycin B to homozygous T$_3$ lines.

For the generation of amiRNA$^{ATG6}$ lines, we utilized *Agrobacterium tumefaciens* strain GV3101 containing the stem-loop structure of the amiRNA$^{ATG6}$ precursor for the transformation of Col plants. The transformation procedure employed was the same as that used for *ATG6-GFP* plants. Positive plants with 50 mg/l kana to homozygous T$_3$ lines.

*UBQ10::ATG6-mCherry × 35S::NPR1-GFP/npr1-2* (*ATG6-mCherry × NPR1-GFP*) was obtained by crossing female *NPR1-GFP/npr1-2* with *ATG6-mCherry*. Screen positive plants with 50 mg/l kana and 30 mg/l hygromycin B. To visualize the localization of NPR1 and ATG6, positive plants with GFP (excitation at 488 nm wavelengths, detection of 500–550 nm wavelengths) and mCherry (excitation at 561 nm wavelengths, detection of 570–650 nm wavelengths) fluorescence were screened through laser scanning confocal microscopy (Zeiss LSM880). To avoid the possibility that the observed fluorescence was due to free mCherry and free GFP, we also verified the presence of ATG6-mCherry and NPR1-GFP in *ATG6-mCherry × NPR1-GFP* plants using western blot experiments. In addition, the levels of NPR1-GFP and free GFP in *ATG6-mCherry × NPR1-GFP* plants were detected before and after SA treatment. Only ~10% of free GFP was detected in *ATG6-mCherry × NPR1-GFP* plants before and after SA treatment (*Figure 2—figure supplement 2*). This also means that the fluorescence signal observed by laser scanning confocal microscopy is dominated by NPR1-GFP, not free GFP. For the identification of *npr1-2*, PCR was performed according to the following primer pairs F: GGATGATTTCTACAGCGACGCT, R: GTAACCATAGCT TA ATGCAGATGGTG. PCR procedure as follows. 95°C 5 min, 95°C 30 s, 55°C 30 s, 72°C 30 s, 72°C 5 min, 30 cycles. The PCR product was then digested by FspI (R0135V, NEB) at 37°C for 10 min (reaction system: 20 µl PCR produces, 3 µl NEB buffer, 0.3 µl FspI, and 6.7 µl H$_2$O) and analyzed by 3% agarose gel electrophoresis (*Cao et al., 1997*; *Chen et al., 2021a*), by the same method plants were screened to T$_3$.

*NPR1-GFP/atg5* was obtained by crossing. For the identification of *atg5*, the triple primer PCR method was performed using RP + LP and LB + RP according to the following primer pairs, LP, AAAGACCACAGAACCCGAAAC. RP, CCAAATTGAATCTTCACCAGG. LBb1.3, ATTTTGCCGATT TCGGAAC. Screen positive plants with 50 mg/l kana, and then positive plants with GFP fluorescence were screened through laser scanning confocal microscopy. Then, in order to exclude the possible fluorescence effects of free GFP, we also used western blots to verify the presence of NPR1-GFP until to T$_3$ homozygous lines.

## Method 3: For *Pst* DC3000/*avrRps4* culture and infiltration

*Pst* DC3000/*avrRps4* was grown in KB medium containing 50 mg/l kana and 25 mg/l rif for 18–24 hr at 28°C, 180 rpm. Then centrifuged at 4000 × *g* for 10 min. The precipitate was washed twice with 10 mM magnesium chloride (MgCl$_2$) and resuspended. The absorbance of the suspension liquid was measured at 600 nm and gradually diluted from 0.8 to 0.02 (for protein levels) or 0.001 (for growth of pathogenic bacteria). Infiltration with *Pst* DC3000/*avrRps4* was performed by pressure infiltration with a 1-ml syringe through the abaxial leaf surface.

## Method 4: Yeast two-hybrid assay

For the interaction of ATG6 with NPR1, the plasmids combinations NPR1-BD and ATG6-AD, NPR1-N-BD and ATG6-AD, NPR1-C-BD and ATG6-AD were, respectively, co-transformed into the yeast strain AH109, according to the Clontech yeast transformation protocol. Co-transformation plasmid combinations NPR1-BD and AD, BD and ATG6-AD, NPR1-N-BD and AD, NPR1-C-BD, SnRK2.8-AD, and BD and AD as negative controls. Co-transformation of NPR1-BD and SnRK2.8-AD used as positive control. Yeast strains were cultivated on SD/-Trp-Leu for 3 days. Pick a fresh single clone and add 50 µl of SD-2 liquid media. Then, yeast strains are gradually diluted in three gradients (10$^{-1}$, 10$^{-2}$, 10$^{-3}$). The yeast was added to a SD/-Trp-Leu-His-Ade to analyze the interaction.

For the interaction of ATGs and NPRs, pGBKT7-NPRs (NPRs-BD) and pGADT7-ATGs (ATGs-AD) were co-transformed into the yeast strain AH109. Yeast strains were cultured on SD/-Trp-Leu for 3 days. Pick a fresh single clone and add it to 50 µl of SD-2 liquid media. Then yeast was added to a SD/-Trp-Leu-His-Ade with different SA concentrations to analyze their interaction.

## Method 5: Prokaryotic protein expression

*E. coli strain* BL21 (DE3) harboring GST, GST-ATG6, GST-SnRK2.8, and NPR1-His was cultured in LB solid medium with 50 mg/l Ampicillin (Amp) for 12–16 hr. Single clones were selected and grown in 1 ml liquid LB medium overnight at 37°C, 150 rpm. Transfer 1 ml of the bacterial solution to 100 ml of LB medium and incubate until $OD_{600}$ = 0.6–1.0. NPR1-His bacterial solution was induced by 1 mM isopropyl β-D-thiogalactoside (IPTG) at 16°C for 24 hr, 150 rpm; GST, GST-ATG6, and GST-SnRK2.8 bacterial solution were induced by 1 mM IPTG at 16°C for 6 hr, 150 rpm. Then it is centrifuged at 8422 × *g* for 10 min at 4°C. The NPR1-His precipitate was added to four to five times the volume of Ni-buffer A (20 mM Tris-HCl pH 7.5, 300 mM NaCl, 15 mM imidazole, 1 mM β-mercaptoethanol). The precipitate of GST, GST-ATG6, and GST-SnRK2.8 was added to four to five times the volume of PBS buffer (10 mM $Na_2HPO_4$, 140 mM NaCl, 2.7 mM KCl, 1.8 mM $KH_2PO_4$, pH 7.4), and added to the cells were dissolved by ultrasound. Centrifuge at 8422 × *g* for 20 min and aspirate the supernatant.

## Method 6: Nuclear and cytoplasmic separation of NPR1-GFP

Samples (0.5 g) of *Arabidopsis* or *Nicotiana benthamiana* were ground well into powder in liquid nitrogen and then added to 1 ml Hondar buffer (2.5% Ficoll 400, 5% DextranT 400, 0.4 M sucrose, 25 mM Tris-HCl pH 7.5, 10 mM $MgCl_2$, 10 mM β-mercaptoethanol) containing freshly configured 0.5 mM PMSF (329-98-6, Sigma-Aldrich, USA), 40 μM MG115 (47480, Sigma-Aldrich, USA), 500× protease inhibitor cocktail (Aprotinin 500 μg/ml, Leupeptin 500 μg/ml, Pepstatin 500 μg/ml) and 5000× phosphatase inhibitor cocktail ($Na_3VO_4$·$12H_2O$ 10 μg/ml, NaF 2.5 μg/ml). The tissue solution was filtered through a 62-μm nylon mesh filter and then a final concentration of 0.5% Triton 100 was added to the filtrate. The mixture was gently mixed and incubated on ice for 15 min. Centrifuged at 1500 × *g* for 5 min at 4°C. The supernatant (cytoplasmic fraction) was carefully transferred to a new centrifuge tube, while the precipitate was resuspended in 1 ml Hondar buffer containing 1% Triton 100. Centrifugated at 100 × *g* for 1 min at 4°C to remove residual cytoplasmic fractions. Then the upper layer was aspirated and centrifuged at 1500 × *g* for 5 min to obtain the nuclear fraction. The nuclear fraction was washed three times with Hondar buffer, and 100 μl of Hondar buffer was added to dissolve it. The protein concentration of the samples was determined using a Bradford microplate reader (INFINITE M PLEX, Tecan). After 10 min denaturation at 75°C, western blot was performed.

## Method 7: Protein degradation analysis

Samples (0.5 g) of *Arabidopsis* (*NPR1-GFP* and *ATG6-mCherry × NPR1-GFP*) or *N. benthamiana* were fully ground in liquid nitrogen and mixed with 500 μl of basal buffer without protease inhibitors (100 mM Tris-HCl pH 7.5, 1 mM EDTA, 150 mM NaCl, 0.5% (vol/vol) Nonidet P-40). The samples were divided equally into six portions and treated at 25°C for 0, 30, 60, 120, and 180 min (*Spoel et al., 2009*). A sample was treated with 100 μM MG115 for 180 min to inhibit the proteasome degradation pathway. One of the portions was treated with 5 μM concanamycin A (Invitrogen, Waltham, MA, USA) or 30 μM Wortmannin (19545-26-7, MedChemExpress, NJ, USA) for 120 min to inhibit autophagy. The protein samples were denatured at 75°C for 10 min. Subsequently, the samples were subjected to SDS–polyacrylamide gel electrophoresis (SDS–PAGE) electrophoresis and analyzed according to western blotting.

## Method 8: Protein extraction and western blotting

Leaves or seedlings (400 mg) were fully ground in liquid nitrogen and homogenized in 400 μl basal buffer (100 mM Tris-HCl pH 7.5, 1 mM EDTA, 150 mM NaCl, 0.5% (vol/vol) Nonidet P-40, 1 mM PMSF). Add 40 μM MG115, 500× protease inhibitor cocktail and 5000× phosphatase inhibitor cocktail to the basal buffer. The samples were incubated on ice for 30 min, with vortexed every 10 min, followed by centrifuged at 10,142 × *g* (TGL16, Cence, Hunan, China) for 15 min at 4°C. The supernatant was transferred to a new 1.5-ml centrifuge tube. The protein concentration of the samples was determined by Bradford using a microplate reader (INFINITE M PLEX, Tecan). Protein was denatured at 100°C for 10 min. NPR1 protein was denatured at 75°C for 10 min.

The sample was subjected to SDS–PAGE (10%) and the gel was transferred to 0.45 μM (>35 kDa) or 0.22 μM (<35 kDa) polyvinylidene fluoride (PVDF) membrane (IPVH00010, Merck Millipore, Germany) for 60 min (>35 kDa) or 30 min (<35 kDa). The membrane was blocked in TBST (Tris-buffered saline with Tween-20) containing 5% dry milk at room temperature for 2 hr. After three washes with TBST, the membrane was incubated with the primary antibody overnight at 4°C. Then, the membranes were washed three times with TBST, and incubated with secondary antibody at room temperature

for 2 hr. Finally, the membrane was washed three times with TBST and the chemiluminescence was imaged using an image analyzer (Tanon-5200, Shanghai, China).

## Method 9: For the treatment of 3-week-old *N. benthamiana*

*Agrobacterium* was initially cultured in LB solid medium (contains appropriate antibiotics). A single clone was then selected and grown in 30 ml of LB liquid medium with the same antibiotics for 16–18 hr at 28°C, 180 rpm. The supernatant was removed by centrifugation and the precipitate was suspended in the infiltration buffer (10 mM MES pH 5.7, 10 mM $MgCl_2$, 150 μM Acetosyringone). The absorbance of the suspension liquid at 600 nm was gradually diluted to $OD_{600} = 0.5–0.8$. The diluted suspension was mixed in a 1:1 ratio. After incubation at 25°C for 1 hr, the mixed *Agrobacterium* was infiltrated into leaves of 3-week-old *N. benthamiana* using a previously described method (*Jiao et al., 2019*).

## Method 10: For the BiFC assay

The *Agrobacterium* GV3101 mixture, containing various combinations of ATG6-cYFP and NPR1-nYFP, NPR1-nYFP and SnRK2.8-cYFP, nYFP and ATG6-cYFP, NPR1-nYFP and cYFP, and nYFP and SnRK2.8-cYFP, was infiltrated into nls-mCherry transgenic tobacco. After 3 days of infiltration, YFP fluorescence was detected in epidermal cells by laser scanning confocal microscopy using an excitation wavelength of 518 nm, detection at 500–550 nm wavelengths. mCherry was detected using an excitation of 561 nm, detection of 570–650 nm wavelengths.

## Method 11: For SINCs-like condensates observation

*Agrobacterium* strain GV3101 harboring NPR1-GFP and mCherry; NPR1-GFP and ATG6-mCherry were mixed in a 1:1 ratio. After incubation at 25°C for 1 hr, the mixed *Agrobacterium* was infiltrated into leaves of 3-week-old *N. benthamiana*. After 2 days, the leaves were soaked in 1 mM SA solution for 24 hr. And then GFP (excitation of 488 nm wavelengths, detection of 500–550 nm wavelengths) fluorescence signals were observed under the laser scanning confocal microscopy.

## Method 12: For growth of *Pst* DC3000/*avrRps4*

A low dose ($OD_{600} = 0.001$) of *Pst* DC3000/*avrRps4* was infiltrated. After 3 days, two small round leaves (8 mm in diameter) were ground into powder in a 1.5-ml centrifuge tube, and then added 500 μl of $MgCl_2$ to dissolve it as the original solution. The original liquid is gradually diluted in six gradients ($10^{-1}$, $10^{-2}$, $10^{-3}$, $10^{-4}$, $10^{-5}$, and $10^{-6}$). *Pst* DC3000/*avrRps4* were spread onto KB solid media (containing 25 mg/l rif and 50 mg/l kana). After 2 days, the colony number was counted according to a previous description (*Wang et al., 2016*; *Lei et al., 2020*).

## Method 13: Real-time quantitative PCR

Total RNA was extracted using Trizol RNA reagent (Invitrogen, 10296-028, Waltham, MA, USA). cDNA was synthesized from 1 μg high-quality total RNA using RT Reagent Kit (TaKaRa, RR047A, Dalian, China). The qPCR was performed on ABI Life QuantStudio 6 using the Low ROX Premixed ChamQ SYBR qPCR Master Mix (Vazyme, Q331-02, Nanjing, China). The qPCR thermal cycles were as follows: 95°C for 30 s, followed by 40 cycles of 95°C for 5 s and 60°C for 34 s. *AtActin2* was used as a control. Each reaction was independently repeated at least three times. Relative expression levels of target genes were calculated using the relative $2^{-\Delta\Delta Ct}$ method (*Zhang et al., 2023*).

## Accession numbers

Sequence data in this article can be found in the GenBank/TAIR databases under the following accession numbers: *Actin2*, At3G18780; *NPR1*, AT1G64280; *ATG6*, AT3G61710; *SnRK2.8*, AT1G78290; *PR1*, AT2G14610; *PR5*, AT1G75040; *ICS1*, AT1G74710.

## Appendix 4

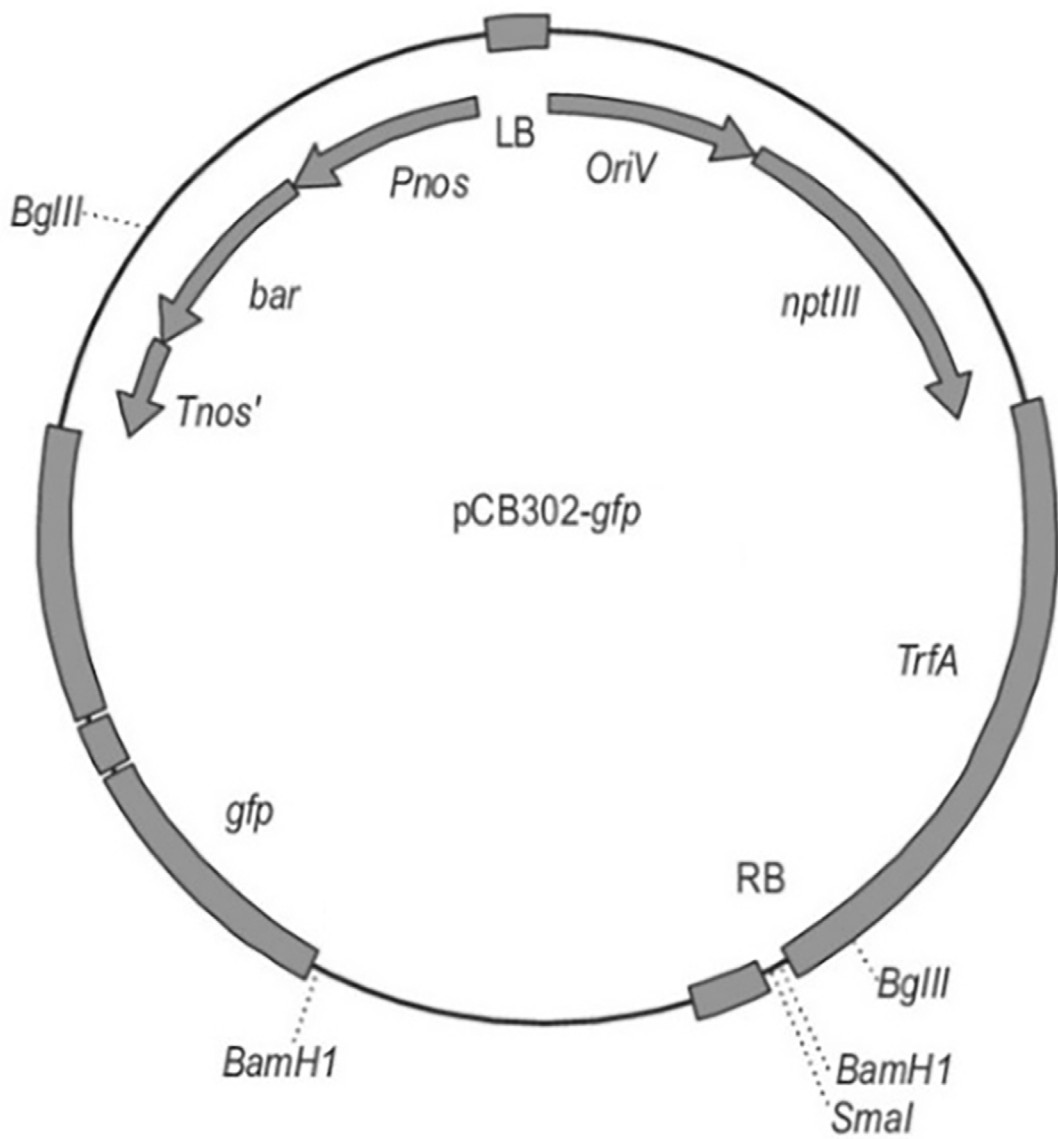

**Appendix 4—figure 1.** Map of pCB302-GFP vector.

Created with SnapGene®

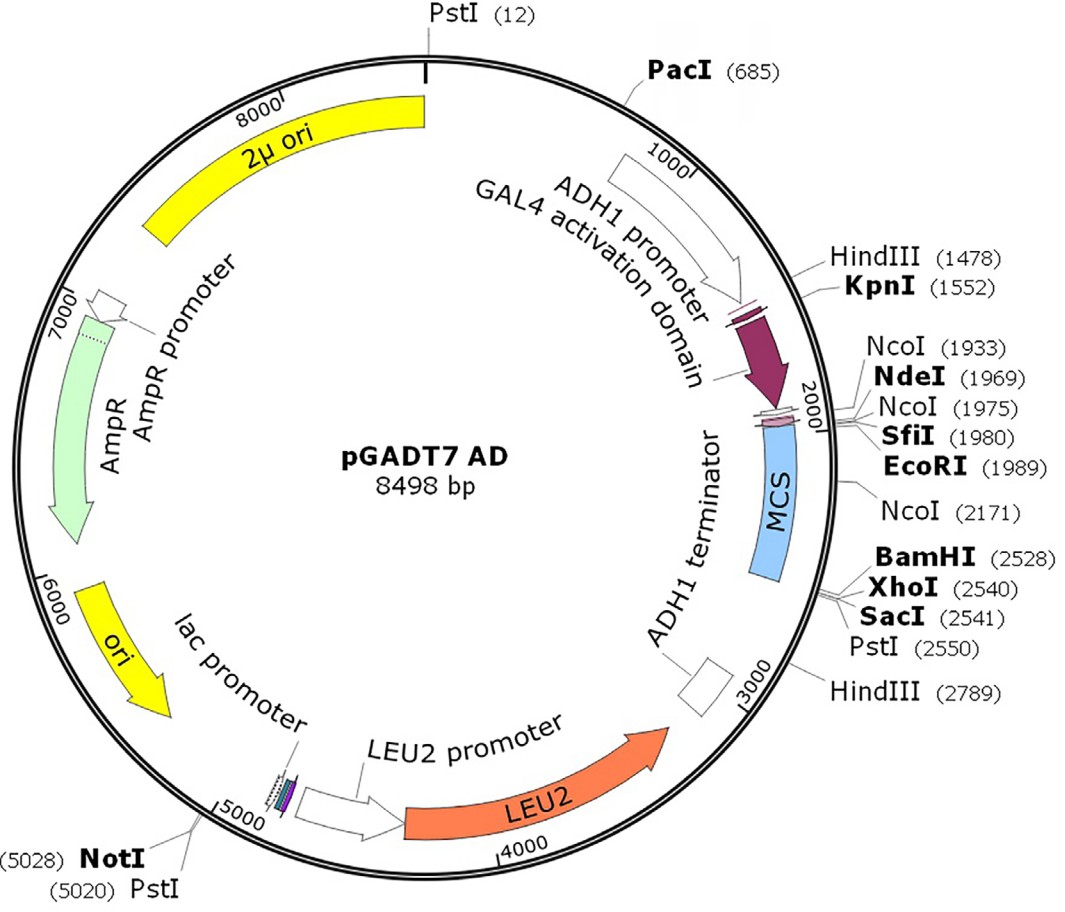

**Appendix 4—figure 2.** Map of pGADT7 AD vector.

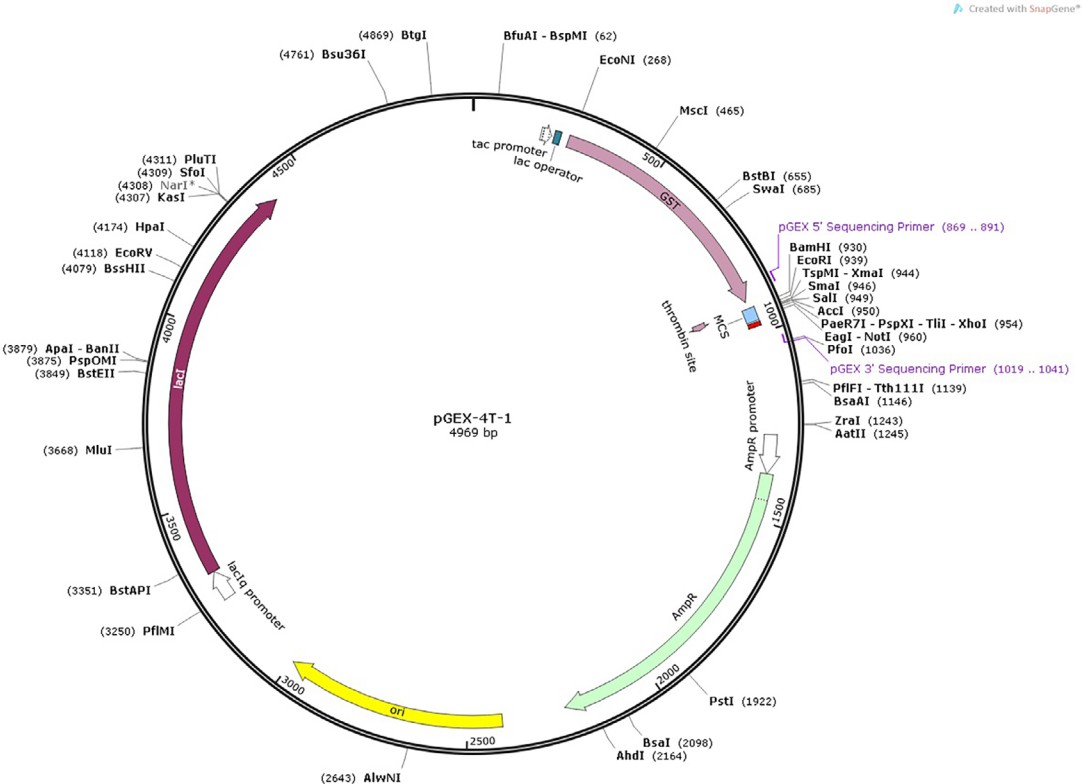

**Appendix 4—figure 3.** Map of pGBKT7 BD vector.

**Appendix 4—figure 4.** Map of pGEX-4T-1 vector.

Created with SnapGene®

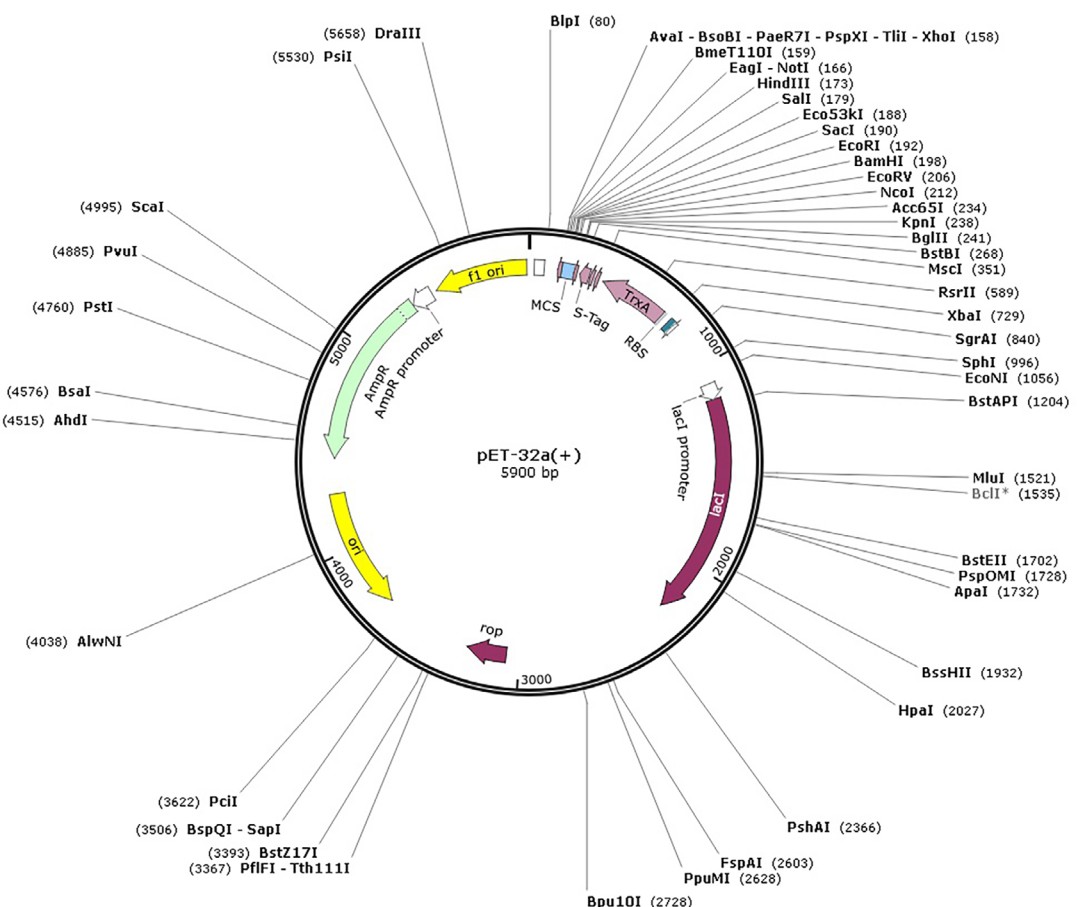

**Appendix 4—figure 5.** Map of pET-32a(+) vector.

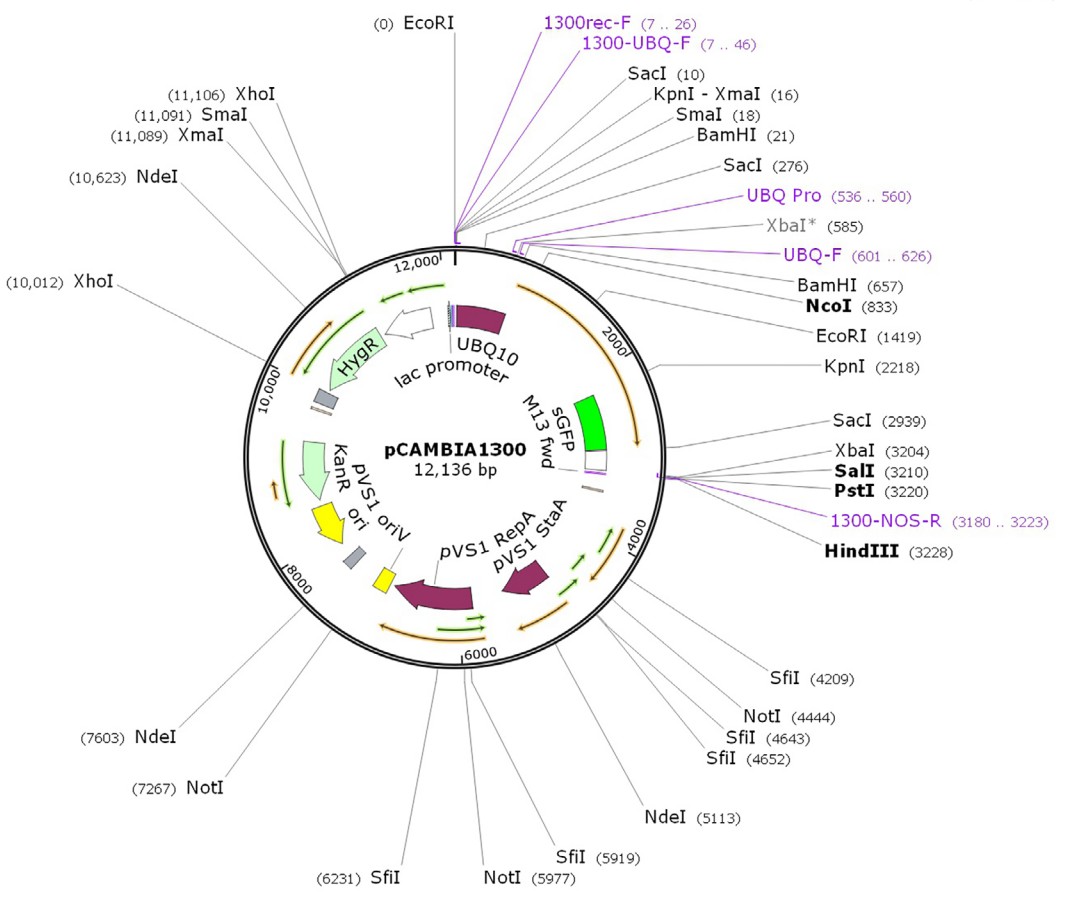

**Appendix 4—figure 6.** Map of pCAMBIA1300 UBQ-sGFP vector.

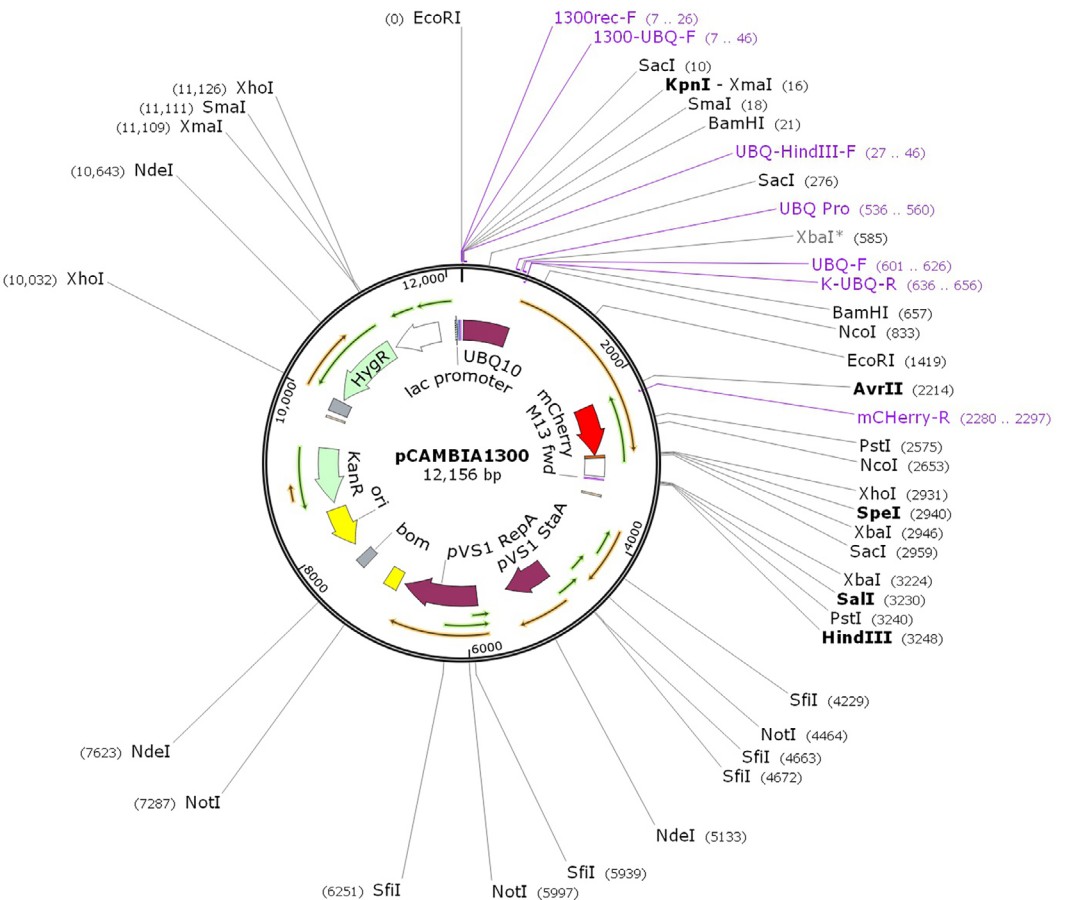

**Appendix 4—figure 7.** Map of pCAMBIA1300 UBQ-mCherry vector.

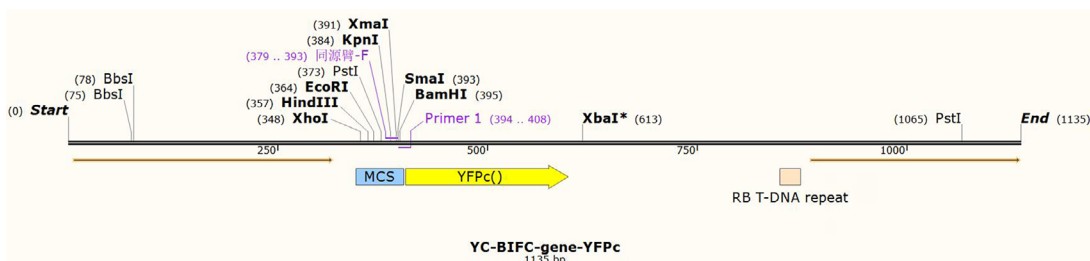

**Appendix 4—figure 8.** Map of YC-BIFC-gene-cYFP vector.

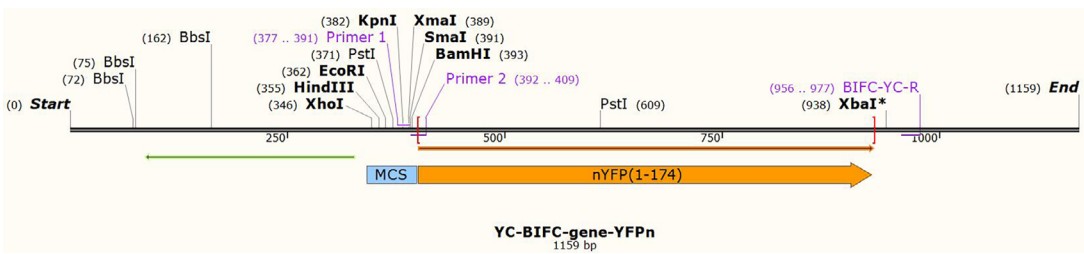

**Appendix 4—figure 9.** Map of YC-BIFC-gene-nYFP vector.

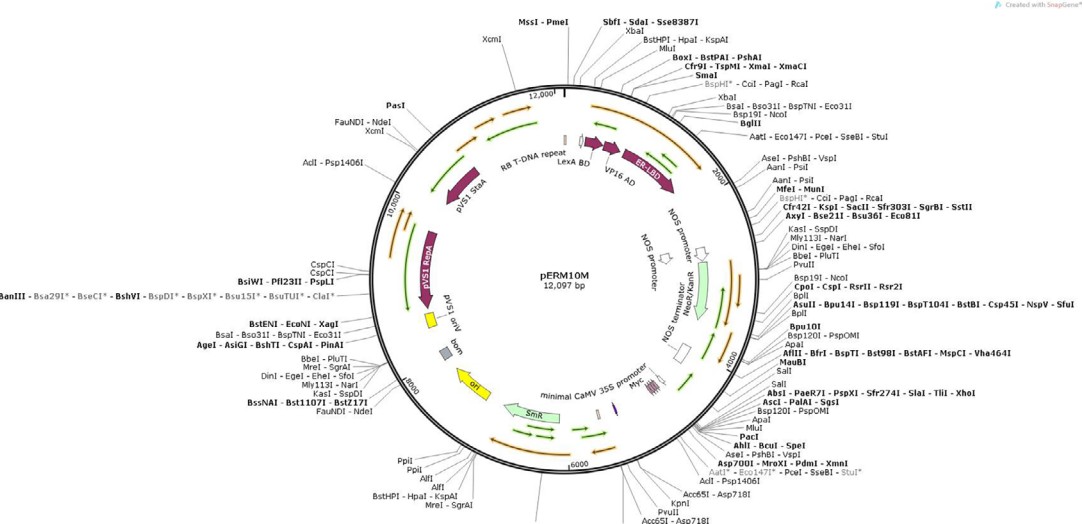

**Appendix 4—figure 10.** Map of pERM10 vector.

