## [Editor Report · eLife Assessment]

This **important** study investigates the role of ATG6 in regulating NPR1, a key protein in the plant immune response. The authors present **compelling** evidence that ATG6 not only interacts with NPR1 in both the cytoplasm and nucleus but also enhances its stability and nuclear accumulation, leading to increased resistance to Pst DC3000/avrRps4 infection in *Arabidopsis thaliana*. The work incorporates a variety of approaches from molecular biology, confocal imaging, and biochemistry, which together strengthen the conclusions.

---

## [Referee Report · Reviewer #1 (Public Review)]

The authors showed that autophagy-related genes are involved in plant immunity by regulating the protein level of the salicylic acid receptor, NPR1.

The experiments are carefully designed and the data is convincing. The authors did a good job of understanding the relationship between ATG6 and NRP1.

Comments on latest version:

The authors have sufficiently addressed all concerns raised, which further enhanced data presentation. No additional concerns were raised.

---

## [Referee Report · Reviewer #2 (Public Review)]

The manuscript by Zhang et al. explores the effect of autophagy regulator ATG6 on NPR1-mediated immunity. The authors propose that ATG6 directly interacts with NPR1 in the nucleus to increase its stability and promote NPR1-dependent immune gene expression and pathogen resistance. This novel role of ATG6 is proposed to be independent of its role in autophagy in the cytoplasm. The authors demonstrate through biochemical analysis that ATG6 interacts with NPR1 in yeast and very weakly in vitro. They further demonstrate using overexpression transgenic plants that in the presence of ATG6-mcherry the stability of NPR1-GFP and its nuclear pool is increased.

Comments on latest version:

The initial apprehensions about statistical oversights and the use of an unclear nuclear marker were fixed. The implementation of the nls-mCherry for nuclear co-localization and additional statistical analyses was done well. However, the functional importance pertaining to cytoplasmic accumulation of the ARG6 protein should ideally be explored in more detail in future studies.

Updated sections:

• Figure 1e: Added statistical analysis and updated with a nuclear marker.

• Line Revisions: Terminology corrections for "infection" instead of "invasion".

• NLS Analysis: Extended alignment and inclusion of conserved domains with predicted NLS (cut-off score: 2.6).

---

## [Author Response]

The following is the authors’ response to the previous reviews.

**Public Reviews:**

**Reviewer #2 (Public Review):**
The manuscript by Zhang et al. explores the effect of autophagy regulator ATG6 on NPR1-mediated immunity. The authors propose that ATG6 directly interacts with NPR1 in the nucleus to increase its stability and promote NPR1-dependent immune gene expression and pathogen resistance. This novel role of ATG6 is proposed to be independent of its role in autophagy in the cytoplasm. The authors demonstrate through biochemical analysis that ATG6 interacts with NPR1 in yeast and very weakly in vitro. They further demonstrate using overexpression transgenic plants that in the presence of ATG6-mcherry the stability of NPR1-GFP and its nuclear pool is increased.Comments on latest version:The term "invasion" has to be replaced with infection, as it doesn't have much meaning to this particular study. I already explained this point in the first review, but authors did not address it throughout the manuscript.

Thank you for your constructive feedback. We have taken your suggestion into account and replaced "invasion" with "infection" in the revised manuscript (Lines 44,45,99,100,298,341,387,415,461,463,464,1002).

In fig. 1e there's no statistical analysis. How can one show measurements from multiple samples without statistical analysis? All the data points have to be shown in the graph and statistics performed. In the arg6-npr1 and snrk-npr1 pairs no nuclear marker is included. How can one know where the nucleus is, particularly in such poor quality low res. images? The nucleus marker has to be included in this analysis and shown. This is an important aspect of the study as nuclear localization of ATG6 is proposed to be essential for its new function.

Thank you for bringing this to our attention. We conducted the BIFC experiments again using *nls-mCherry* transgenic tobacco, which yielded clearer images. The results clearly demonstrate that ATG6 interacts with NPR1 in both the cytoplasm and nucleus. YFP signaling in the nucleus co-localizes with nls-mCherry (a nuclear localization mark). SnRK2.8 was employed as a positive control for NPR1 interaction." Relative fluorescence intensity of YFP were analyzed using image J software, n = 15 independent images were analyzed to quantify YFP fluorescence. All data points are displayed in the image, and we also conducted a Student's t-test analysis. We have incorporated these results into the revised manuscript (Fig 1d and e).

Co-localization provided in the fig. S2 cannot complement this analysis, particularly since no cytoplasmic fraction is present for NPR1-GFP in fig. S2.

Thank you for your observation. We repeated the experiment and confirmed that NPR1 and ATG6 co-localize in both the nucleus and cytoplasm. The image in Figure S2 has been updated accordingly.

In the alignment in fig 2c, it is not explained what are the species the atg6 is taken from. The predicted NLS has to be shown in the context of either the entire protein sequence alignment or at least individual domain alignment with the indication of conserved residues (consensus). They have to include more species in the analysis, instead of including 3 proteins from a single species. Also, the predicted NLS in atg6 doesn't really have the classical type architecture, which might be an indication that it is a weak NLS, consistent with the fact that the protein has significant cytoplasmic accumulation. They also need to provide the NLS prediction cut-off score, as this parameter is a measure of NLS strength.Line 150: the NLS sequence "FLKEKKKKK" is a wrong sequence.

Thank you for your suggestion. In both plants and animals, proteins are transported to the nucleus via specific nuclear localization signals (NLSs), which are typically characterized by short stretches of basic amino acids (Dingwall and Laskey, 1991, Raikhel, 1992, Nigg, 1997). Following your recommendation, we re-predicted potential NLS sequences in the ATG6 protein using NLSExplorer (http://www.csbio.sjtu.edu.cn/bioinf/NLSExplorer). Although we did not identify a classical monopartite NLS, we discovered a bipartite NLS similar to the consensus bipartite sequence (KRX_(10-12)_K(KR)(KR)) (Kosugi *et al.*, 2009)in the carboxy-terminal region (475-517 aa) of ATG6, with a cut-off score of 2.6. These findings are consistent with substantial accumulation of ATG6 in the cytoplasm and minimal accumulation in the nucleus. Additionally, our comparison of ATG6 C-terminal sequences across several species, including *Microthlaspi erraticum*, *Capsella rubella*, *Brassica carinata*, *Camelina sativa*, *Theobroma cacao*, *Brassica rapa*, *Eutrema salsugineum*, *Raphanus sativus*, *Hirschfeldia incana* and *Brassica napus*, sequence comparison indicates that this bipartite NLS is relatively conserved. We have incorporated these results into the revised manuscript (lines 450-160).

In fig. 3d no explanation for the error bars is included, and what type of statistical analysis is performed is not explained.

Thank you for bringing this to our attention. In Figure 3d, a Student's t-test was conducted to analyze the data. The mean and standard deviation were calculated from three biological replicates, and the relevant description has been included in the figure notes.

Reference

Dingwall, C. and Laskey, R.A. (1991) Nuclear targeting sequences--a consensus? *Trends Biochem Sci*, 16, 478-481.

Kosugi, S., Hasebe, M., Matsumura, N., Takashima, H., Miyamoto-Sato, E., Tomita, M. and Yanagawa, H. (2009) Six classes of nuclear localization signals specific to different binding grooves of importin alpha. *J Biol Chem*, 284, 478-485.

Nigg, E.A. (1997) Nucleocytoplasmic transport: signals, mechanisms and regulation. *Nature*, 386, 779-787.

Raikhel, N. (1992) Nuclear targeting in plants. *Plant Physiol*, 100, 1627-1632.